# Formin-like 1β phosphorylation at S1086 is necessary for secretory polarized traffic of exosomes at the immune synapse in Jurkat T lymphocytes

Javier Ruiz-Navarro[1†], Sara Fernández-Hermira[1†], Irene Sanz-Fernández[1†], Pablo Barbeito[1†], Alfonso Navarro-Zapata[2,3], Antonio Pérez-Martínez[2,3,4,5], Francesc R Garcia-Gonzalo[1,6,7†], Víctor Calvo[1†], Manuel Izquierdo Pastor[1*†]

[1]Instituto de Investigaciones Biomédicas Sols-Morreale (IIBM), CSIC-UAM, Madrid, Spain; [2]Translational Research in Pediatric Oncology, Hematopoietic Transplantation and Cell Therapy, IdiPAZ, La Paz University Hospital, Madrid, Spain; [3]Pediatric Onco-Hematology Clinical Research Unit, Spanish National Cancer Center (CNIO), Madrid, Spain; [4]Department of Pediatric Hemato-Oncology, La Paz University Hospital, Madrid, Spain; [5]Pediatric Department, Autonomous University of Madrid, Madrid, Spain; [6]CIBER de Enfermedades Raras (CIBERER), Instituto de Salud Carlos III (ISCIII), Madrid, Spain; [7]Instituto de Investigación Sanitaria del Hospital Universitario La Paz (IdiPAZ), Madrid, Spain

*For correspondence: manuel.izquierdo@inv.uam.es

Present address: †Departamento de Bioquímica, Facultad de Medicina, UAM, Madrid, Spain

Competing interest: The authors declare that no competing interests exist.

## eLife Assessment

This **important** study uses the Jurkat T cell model to study the role of Formin-like 1 β phosphorylation at S1086 on actin dynamics and exosome release at the immunological synapse. The evidence supporting these findings is **compelling** within the framework of the Jurkat model. As the Jurkat model is known to have a bias toward formin-mediated actin filament formation at the expense of Arp2/3-mediated branched F-actin foci observed in primary T cells, it will be beneficial in the future to confirm major findings in primary T cells.

**Abstract** We analyzed here how formin-like 1 β (FMNL1β), an actin cytoskeleton-regulatory protein, regulates microtubule-organizing center (MTOC) and multivesicular bodies (MVB) polarization and exosome secretion at an immune synapse (IS) model in a phosphorylation-dependent manner. IS formation was associated with transient recruitment of FMNL1β to the IS, which was independent of protein kinase C δ (PKCδ). Simultaneous RNA interference of all FMNL1 isoforms prevented MTOC/MVB polarization and exosome secretion, which were restored by FMNL1βWT expression. However, expression of the non-phosphorylatable mutant FMNL1βS1086A did not restore neither MTOC/MVB polarization nor exosome secretion to control levels, supporting the crucial role of S1086 phosphorylation in MTOC/MVB polarization and exosome secretion. In contrast, the phosphomimetic mutant, FMNL1βS1086D, restored MTOC/MVB polarization and exosome secretion. Conversely, FMNL1βS1086D mutant did not recover the deficient MTOC/MVB polarization occurring in PKCδ-interfered clones, indicating that S1086 FMNL1β phosphorylation alone is not sufficient for MTOC/MVB polarization and exosome secretion. FMNL1 interference inhibited the depletion of F-actin at the central region of the immune synapse (cIS), which is necessary for MTOC/MVB polarization. FMNL1βWT and FMNL1βS1086D, but not FMNL1βS1086A expression, restored F-actin depletion at the cIS. Thus, actin cytoskeleton reorganization at the IS

underlies the effects of all these FMNL1β variants on polarized secretory traffic. FMNL1 was found in the IS made by primary T lymphocytes, both in T cell receptor (TCR) and chimeric antigen receptor (CAR)-evoked synapses. Taken together, these results point out a crucial role of S1086 phosphorylation in FMNL1β activation, leading to cortical actin reorganization and subsequent control of MTOC/MVB polarization and exosome secretion.

## Introduction

T cell receptor (TCR) stimulation by antigen bound to major histocompatibility complex (MHC) molecules on the surface of an antigen-presenting cell (APC) induces the formation of the immune synapse (IS). The IS is a specialized cell-cell interface contact area that provides a signaling platform for integration of signals leading to intercellular information exchange, in order to ensure efficient TCR signal transduction, T cell activation, and the proper execution of diverse T lymphocyte effector functions (*Dustin and Choudhuri, 2016*). Comprised among these functions, IS formation triggers the convergence of T lymphocyte secretory vesicles, including multivesicular bodies (MVB) (*Alonso et al., 2011*; *Calvo and Izquierdo, 2021*), toward the microtubule-organizing center (MTOC) and the polarization of MTOC together with secretory vesicles to the IS (*de la Roche et al., 2016*; *Huse, 2012*). The canonical bullseye structure of a mature IS comprises three concentric areas referred to as supramolecular activation clusters (SMACs) that can be discerned based on the specific segregation of TCR, integrins, and co-stimulatory molecules (*Monks et al., 1998*). The central supramolecular activation cluster (cSMAC) is enriched in ligand-bound TCR and associated signaling molecules, the peripheral SMAC (pSMAC) in adhesion molecules such as LFA-1, and the outer distal SMAC (dSMAC) in filamentous actin (F-actin) and CD45 (*Hammer et al., 2019*; *Blumenthal and Burkhardt, 2020*). IS assembly and subsequent secretory responses are coordinated by both actin and microtubule cytoskeletons, whose interplay regulates the polarized secretory response toward the IS (*Ritter et al., 2013*; *Hammer et al., 2019*). IS formation is associated with an initial increase in cortical actin at the IS (*Billadeau et al., 2007*), followed by a decrease in cortical actin density at the central region of the immune synapse (cIS) including the cSMAC, that contains the secretory domain (*Griffiths et al., 2010*; *Ritter et al., 2015*). The partial depletion of F-actin at the cIS has been proposed, apart from allowing the focusing and docking of secretory vesicles for secretion (*Billadeau et al., 2007*), to initiate key subsequent events leading to MTOC polarization and delivery of secretory vesicles, such as lytic granules in cytotoxic T lymphocytes (CTL) (*Ritter et al., 2015*) and cytokine-containing secretory vesicles in T-helper (Th) lymphocytes (*Chemin et al., 2012*), to the secretory domain of the IS. The CTL degranulation of diverse secretory lytic granules with MVB structure results in the secretion to the synaptic cleft of Fas ligand (FasL)-containing exosomes (*Peters et al., 1989*; *Peters et al., 1991*; *Martínez-Lorenzo et al., 1999*; *Alonso et al., 2005*; *Alonso et al., 2011*; *Mazzeo et al., 2016*). These exosomes, along with perforin and granzymes, which are secreted in both soluble and nanoparticulate form in the so-called supramolecular attack particles (*Bálint et al., 2020*; *Cassioli and Baldari, 2022*), lead to the induction of target cell apoptosis. In addition, FasL-containing exosomes have been involved in autocrine FasL/Fas-dependent, activation-induced cell death (AICD) produced upon TCR triggering (*Martínez-Lorenzo et al., 1999*; *Monleón et al., 2001*; *Alonso et al., 2005*; *Calvo and Izquierdo, 2020*), an important immunoregulatory event involved in the downregulation of T cell immune responses (*Krammer et al., 2007*; *Nagata and Suda, 1995*).

We have previously described that cortical actin reorganization at the IS plays an important role in MVB polarized traffic leading to exosome secretion in Th lymphocytes (*Herranz et al., 2019*; *Bello-Gamboa et al., 2020*; *Calvo and Izquierdo, 2021*). With respect to the molecular cues controlling this process, we have shown that TCR-stimulated protein kinase C δ (PKCδ) regulates cortical actin reorganization at the IS, thereby controlling MTOC/MVB polarization and ultimately leading to exosome secretion at the IS and AICD in Th lymphocytes (*Herranz et al., 2019*). Moreover, PKCδ is necessary for the polarization of lytic granules and the induction of cytotoxicity by mouse CTL (*Ma et al., 2008*; *Ma et al., 2007*), which supports a general role of PKCδ in secretory traffic leading to apoptosis in T lymphocytes. Besides PKCδ, several actin cytoskeleton regulators, including the formin FMNL1 and Diaphanous-1 (Dia1), also regulate MTOC polarization (*Gomez et al., 2007*; *Kühn and Geyer, 2014*; *Kumari et al., 2014*). In this context, we have shown both PKCδ-dependent phosphorylation of FMNL1 at the IS and PKCδ-dependent F-actin clearing at the cIS, which participate in MTOC/

MVB polarization leading to exosome secretion in CD4[+] Jurkat T lymphocytes forming IS (*Herranz et al., 2019*; *Bello-Gamboa et al., 2020*; *Calvo and Izquierdo, 2021*). However, a formal connection between FMNL1 phosphorylation, F-actin regulation, and MVB polarization/exosome secretion at the IS has not been established yet.

Regarding potential formin regulatory pathways, we have shown that FMNL1, but not Dia1, is strongly phosphorylated in T lymphocytes by PKCδ activators such as phorbol myristate acetate (PMA), but also upon TCR stimulation via an anti-TCR agonist as well as IS formation (*Bello-Gamboa et al., 2020*). FMNL1 phosphorylation was inhibited in PKCδ-interfered T lymphocytes (*Bello-Gamboa et al., 2020*), which is associated with a deficient MTOC polarization toward the IS (*Herranz et al., 2019*), supporting a PKCδ role in both FMNL1 phosphorylation and MTOC polarization (*Herranz et al., 2019*; *Bello-Gamboa et al., 2020*). Interestingly, FMNL2, a related formin which exhibits a high homology with FMNL1, is phosphorylated by PKCα and, to a lower extent, by PKCδ, at S1072 (*Wang et al., 2015*). This phosphorylation reverses FMNL2 autoinhibition mediated by interaction of N-terminal Diaphanous inhibitory domain (DID) with the C-terminal Diaphanous autoinhibitory domain (DAD), resulting in increased F-actin assembly, β1-integrin endocytosis, invasive motility (*Wang et al., 2015*), and filopodia elongation (*Lorenzen et al., 2023*). In this regard, recent evidence suggests that the mutation of a particular residue in FMNL2 impacting the DID-DAD interaction may result in functional implications for the characteristic actin-regulating activity of FMNL2, specifically in the context of podosome formation in macrophages (*Trefzer et al., 2021*). Out of the three FMNL1 isoforms (α, β, and γ) present in T lymphocytes and Jurkat cells (*Colón-Franco et al., 2011*), S1086 in FMNL1β is

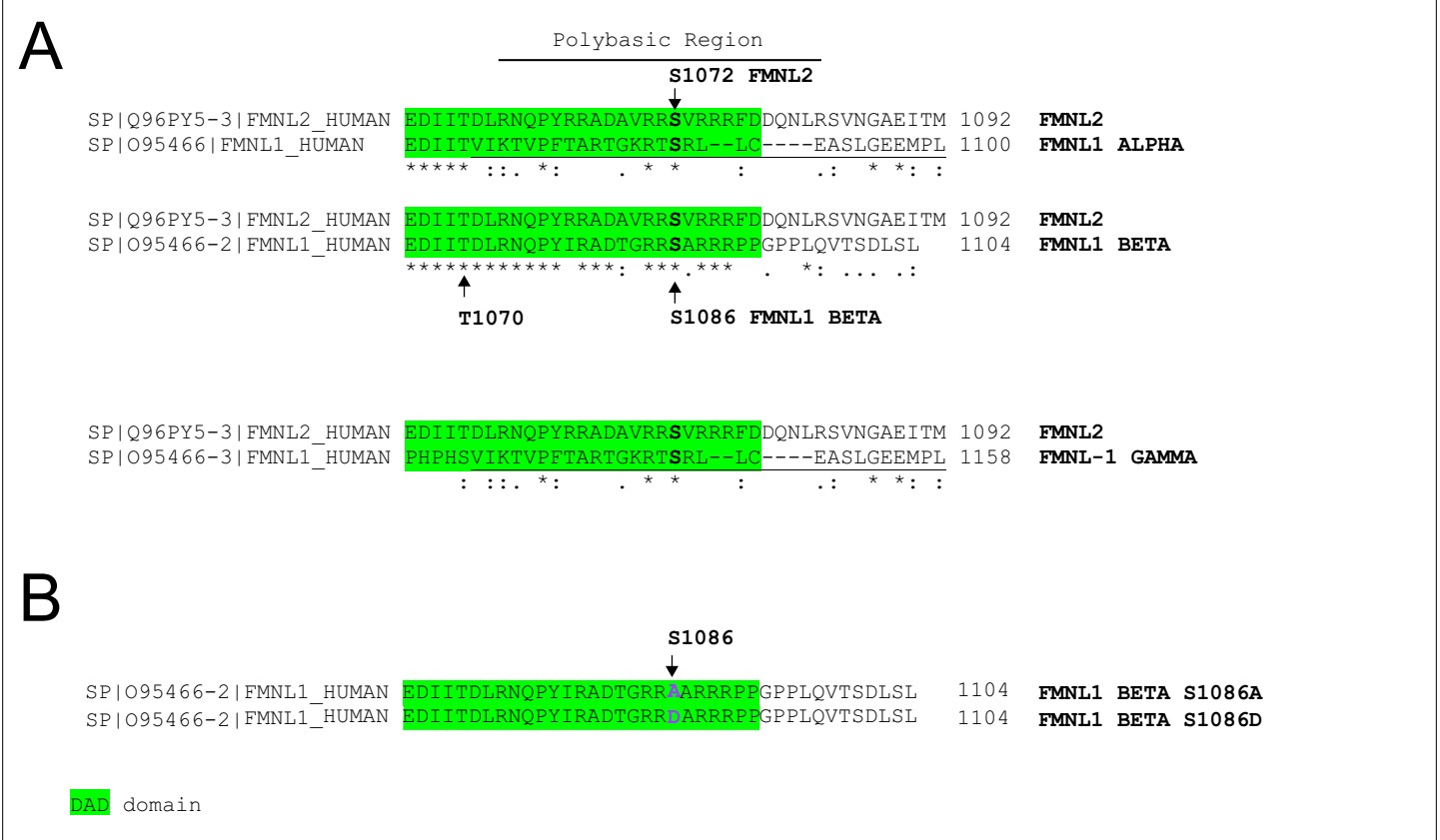

**Figure 1.** C-terminal alignment of FMNL1 isoforms. S1086 of FMNL1β shows a high degree of homology with protein kinase C (PKC)-phosphorylated S1072 in FMNL2. (**A**) Amino acid sequences of the C-termini of FMNL2, FMNL1α, and FMNL1β, as well as FMNL1γ containing a C-terminal intron retention, and sharing the final C-terminal amino acids with FMNL1α (shown underlined). The three FMNL1 isoforms share identical sequence (100% identity) from amino acid residue 1 to T1070, and diverge in the C-terminal region, which includes the Diaphanous autoinhibitory domain (DAD) autoinhibitory domain (*Han et al., 2009*). The DAD domain sequence, which is responsible for autoinhibition in the murine homolog, is highlighted in green (*Han et al., 2009*), and the identical C-terminal amino acids shared by FMNL1α and FMNL1γ are underlined. The arginine-rich, polybasic region common to other formins such as FMNL2 (*Wang et al., 2015*), FHOD3 (*Zhou et al., 2017*), and FHOD1 (*Takeya et al., 2008*) (see Discussion) is also shown overlined. (**B**) C-terminal amino acid sequences of the two different point mutations at S1086 in FMNL1β used in this study.

surrounded by a sequence displaying high homology to the one surrounding S1072 in FMNL2 (*Wang et al., 2015*; *Bello-Gamboa et al., 2020*; *Figure 1*). Moreover, we have shown that FMNL1β is the only FMNL1 isoform phosphorylated upon IS formation and capable of recovering MTOC polarization when it was re-expressed in cells simultaneously interfered for the three FMNL1 isoforms (*Bello-Gamboa et al., 2020*). Thus, we hypothesize that IS-induced, PKCδ-dependent phosphorylation at S1086, a residue located within FMNL1β DAD, may release the autoinhibition mediated by DID/DAD interaction, thereby leading to the activation of FMNL1β. Consequently, this activation would mediate the F-actin reorganization at the IS and the MTOC/MVB polarization, thereby enabling the subsequent exosome secretion. Here, we report that FMNL1β phosphorylation at S1086 regulates F-actin reorganization leading to MTOC/MVB polarization and exosome secretion.

## Results

### FMNL1 interference and YFP-FMNL1β variants expression

According to our previous data (*Bello-Gamboa et al., 2020*), FMNL1β plays a crucial role in MTOC polarization toward the IS. Moreover, we have also shown that FMNL1β is strongly phosphorylated in T lymphocytes by PKCδ activators such as PMA, as well as by TCR stimulation and upon IS formation, which could potentially be related to FMNL1β activation (*Bello-Gamboa et al., 2020*). As previously stated, FMNL2 becomes active upon PKCα phosphorylation at S1072, enhancing F-actin assembly (*Wang et al., 2015*; *Lorenzen et al., 2023*). In FMNL1β, S1086 in arginine-rich DAD (labeled in *Figure 1*) is surrounded by a sequence displaying high similarity to that around S1072 in FMNL2 (*Han et al., 2009*; *Wang et al., 2015*; *Bello-Gamboa et al., 2020*; *Figure 1A*). Thus, we hypothesized that phosphorylation of FMNL1β at S1086 may release DID-DAD autoinhibition, thereby activating FMNL1β and regulating MTOC polarization. To address this point, we introduced S1086A and S1086D mutations in the shFMNL1-HA-YFP-FMNL1β construct (*Colón-Franco et al., 2011*; *Bello-Gamboa et al., 2020*), which act as non-phosphorylatable and phosphomimetic residues, respectively (*Figure 1B*). These resulting bi-cistronic vectors produce both FMNL1 interference and the corresponding YFP-FMNL1β variant expression (*Colón-Franco et al., 2011*), acting as useful tools to analyze the role of FMNL1β in IS formation during transient expression experiments (*Bello-Gamboa et al., 2020*). To confirm the previously described efficacy of these vectors (*Lorenzen et al., 2023*; *Bello-Gamboa et al., 2020*), we first analyzed endogenous FMNL1 interference and YFP-FMNL1β variants expression in C3 Jurkat clone transfected with the different vectors at the single cell level by immunofluorescence, using an anti-FMNL1 antibody that recognizes all its isoforms. This analysis was performed in transfected Jurkat cells forming synapses, to rule out any potential effect of IS formation and subsequent T lymphocyte activation on both endogenous FMNL1 interference and YFP-FMNL1β variants expression. As seen in *Figure 2A*, C3 cells transfected with the FMNL1-interfering plasmid shFMNL1-YFP (YFP$^+$ cells) (second row) showed very low anti-FMNL1 signal when compared with non-transfected cells in the same preparation or control YFP$^-$ cells (first row). In contrast, cells transfected with YFP-FMNL1βWT, YFP-FMNL1βS1086A, or YFP-FMNL1βS1086D (rows 3–5, respectively) showed higher anti-FMNL1 fluorescence signals when compared to non-transfected cells in the same microscopy field (*Figure 2A*). *Figure 2—figure supplement 1B* shows, by single cell image analysis, that the mean fluorescence intensity (MFI) of the anti-FMNL1 in shFMNL1-YFP-expressing cells is extremely low when compared to the control YFP$^-$ group, whereas it is much higher in YFP-FMNL1βWT, YFP-FMNL1βS1086A, or YFP-FMNL1βS1086D-expressing cells. As shown in *Figure 2—figure supplement 1A*, anti-FMNL1 MFI correlated with YFP-FMNL1β MFI in YFP-FMNL1βWT, YFP-FMNL1βS1086A, or YFP-FMNL1βS1086D-expressing cells. To further verify this data, lysates from bulk populations of transfected cells were analyzed by western blot (WB) using the same anti-FMNL1 antibody. The apparent molecular weights (MW) of the bands observed in the WB analysis were compatible with the predicted MW of endogenous FMNL1 (150 kDa) and the chimeric YFP-FMNL1β variants (150+30 kDa) (*Figure 2B*). The endogenous FMNL1 (150 kDa band) levels in lysates from cells transfected with YFP-FMNL1βWT, YFP-FMNL1βS1086A, or YFP-FMNL1βS1086D were lower than in control untransfected cells (*Figure 2B*). The reduction of endogenous FMNL1 expression, as assessed by WB (between 40% and 80% reduction relative to control YFP$^-$ cells), was apparently lower than that observed by single cell imaging analysis (compare panel B in *Figure 2* with *Figure 2—figure supplement 1B*). This is

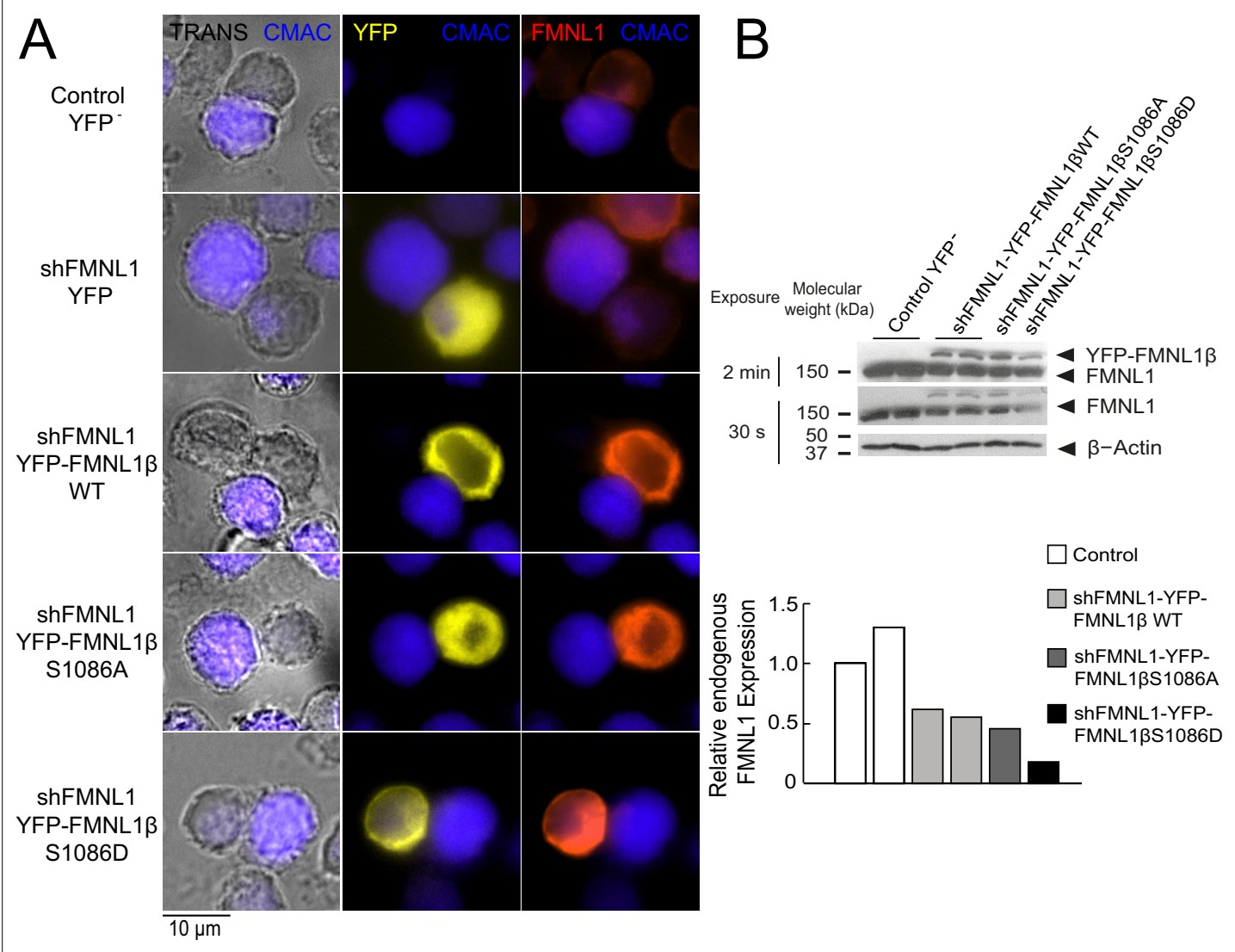

**Figure 2.** Expression of YFP-FMNL1β S1086 variants in FMNL1-silenced cells. (**A**) C3 control clone was untransfected (Control YFP⁻) (first row) or transfected with FMNL1-interfering (shFMNL1-HA-YFP) (second row), or FMNL1-interfering expressing interference-resistant YFP-FMNL1βWT (shFMNL1-HA-YFP-FMNL1βWT) (third row), YFP-FMNL1βS1086A (fourth row), or YFP-FMNL1βS1086D (fifth row) constructs. Subsequently, cells were challenged with CMAC-labeled SEE-pulsed Raji cells (blue) for 1 hr, fixed, stained with anti-FMNL1, and imaged by epifluorescence microscopy. Representative maximum intensity projection (MIP) images of merged transmittance (TRANS), CMAC (blue), YFP (yellow), and anti-FMNL1 (red) channels are indicated for the different cell groups. (**B**) Cell lysates corresponding to the indicated cell groups were analyzed by western blot (WB) developed with anti-FMNL1 antibody (two different expositions). The lower bar graph depicts the WB quantification showing the endogenous FMNL1 expression in the different cell groups relative to control untransfected (YFP⁻) cells. Results are representative of data from several independent experiments (n=3) with similar results.

The online version of this article includes the following source data and figure supplement(s) for figure 2:

**Source data 1.** Original, uncropped WB corresponding to panel B in *Figure 2* in PDF format.

**Source data 2.** Original, uncropped WB corresponding to panel B in *Figure 2* in TIF format.

**Figure supplement 1.** Expression of YFP-FMNL1β constructs and correlation between YFP-FMNL1β variants expression and the anti-FMNL1 antibody signal.

not surprising, since the transfection efficiency was relatively low (20–50%) and thus cell populations analyzed by WB contained transfected as well as untransfected cells.

## YFP-FMNL1βWT is phosphorylated upon PKC activation but neither YFP-FMNL1βS1086A nor YFP-S1086D are

Once the different YFP-FMNL1β variants were expressed, we analyzed the ability of a PKCδ activator to induce their phosphorylation. To this end, we stimulated C3 clone transfected with the different

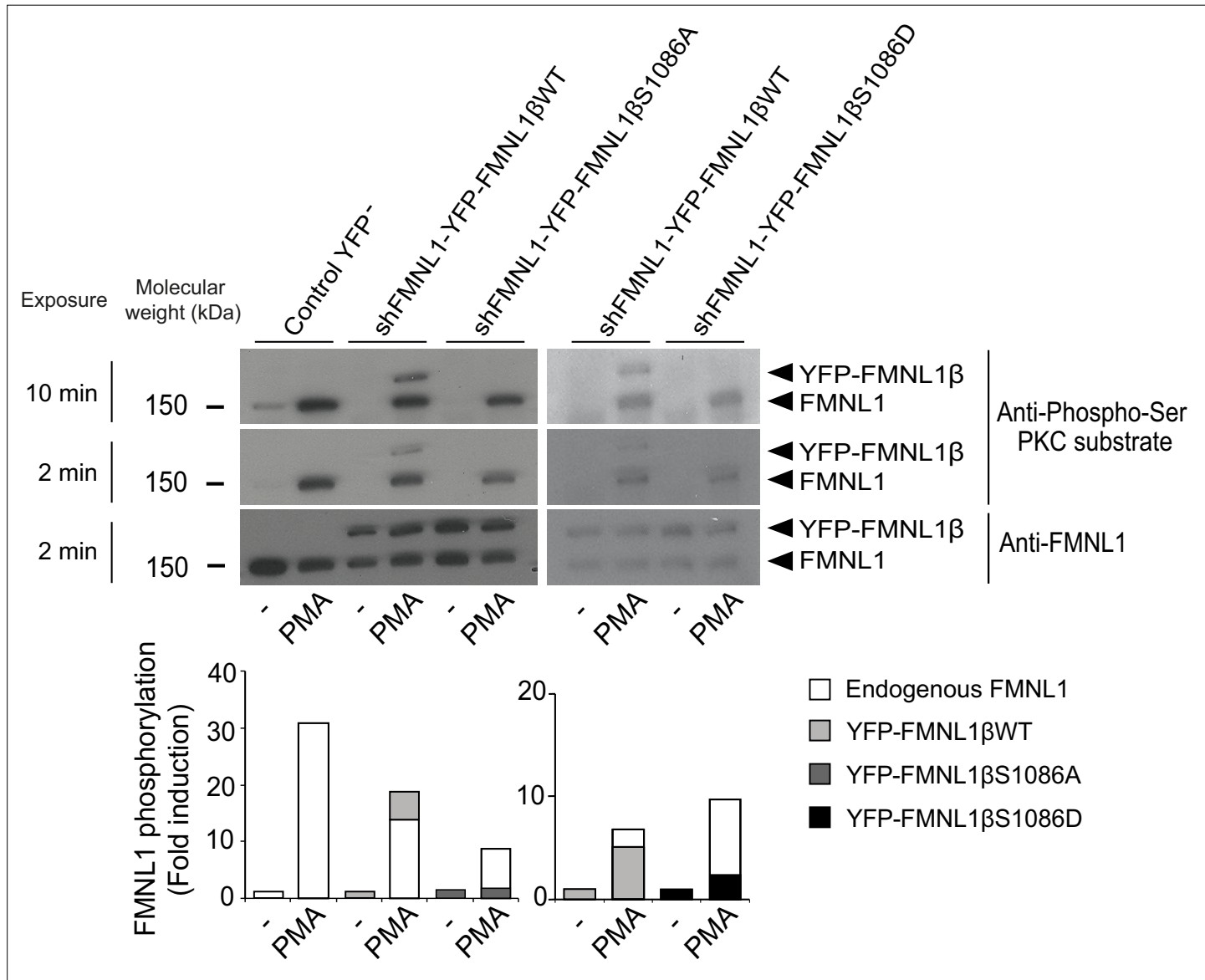

**Figure 3.** S1086 in FMNL1β is phosphorylated upon protein kinase C (PKC) activation. C3 control clone was unstransfected (Control YFP⁻) or transfected with either FMNL1-interfering expressing interference-resistant YFP-FMNL1βWT (shFMNL1-HA-YFP-FMNL1βWT), YFP-FMNL1βS1086A, or YFP-FMNL1βS1086D constructs. Subsequently, cells were stimulated or not (-) with PKCδ activator phorbol myristate acetate (PMA) for 30 min. The different cell groups were lysed and immunoprecipitated with anti-FMNL1. These immunoprecipitates (IPs) were analyzed by western blot (WB), first with anti-Phospho-Ser PKC substrate antibody (two different expositions) and then reprobed with anti-FMNL1 to normalize Phospho-Ser PKC substrate signal for FMNL1 protein levels. The lower graph represents the normalized fold induction of phosphorylation of the different FMNL1 variants. Results are representative of data from several independent experiments (n=3) with similar results.

The online version of this article includes the following source data for figure 3:

**Source data 1.** Original, uncropped WBs corresponding to the upper panel in *Figure 3* in PDF format.

**Source data 2.** Original, uncropped WBs corresponding to the upper panel in *Figure 3* in TIF format.

constructs with the PKCδ activator PMA, which has been shown to induce strong FMNL1 phosphorylation (*Bello-Gamboa et al., 2020*). Subsequently, we immunoprecipitated FMNL1 with an anti-FMNL1 that recognizes all FMNL1 isoforms (*Bello-Gamboa et al., 2020*) and analyzed these immunoprecipitates (IPs) by WB with anti-phospho-Ser PKC substrate (*Wang et al., 2015*; *Figure 3*). In all the IPs from PMA-stimulated cells, anti-phospho-Ser PKC substrate detected a band corresponding to endogenous FMNL1 (150 kDa), which was 8- to 30-fold more intense than in IPs from unstimulated cells (*Figure 3*, lower graph). In IPs from PMA-stimulated cells expressing YFP-FMNL1βWT, anti-phospho-Ser PKC substrate detected an additional band of the predicted MW (150+30 kDa), which was also recognized by anti-FMNL1 (*Figure 3*). However, in IPs from PMA-stimulated cells expressing YFP-FMNL1βS1086A or S1086D no apparent phosphorylation of these FMNL1β chimeras could be detected (*Figure 3*). Thus, substitution of S1086 by a non-phosphorylatable or a phosphomimetic residue fully abolished PKCδ activator-induced phosphorylation of FMNL1β, as detected by anti-phospho-Ser PKC substrate, confirming our hypothesis that S1086 is indeed the residue that is phosphorylated in FMNL1β upon PKC activation.

## S1086 phosphorylation of FMNL1β is crucial for MTOC/MVB polarization toward the IS

Next, we wanted to assess whether S1086 phosphorylation was related to FMNL1β regulation of MTOC/MVB polarization toward the IS. To this end, untransfected (control YFP⁻) C3 cells, or transfected with the plasmids interfering all FMNL1 isoforms and expressing YFP (shFMNL1-YFP), YFP-FMNL1βWT, YFP-FMNL1βS1086A, or YFP-FMNL1βS1086D (*Colón-Franco et al., 2011*) were challenged with SEE-pulsed Raji cells (blue). Next, we quantified MTOC/MVB polarization index (PI) in fixed synapses formed by the mentioned cell groups by epifluorescence microscopy and deconvolution, as previously described (*Obino et al., 2017*; *Herranz et al., 2019*; *Bello-Gamboa et al., 2020*; *Figure 4—figure supplement 1A*, *Figure 4*). In the first series of experiments, average MTOC PI of untransfected (Control YFP⁻) C3 cells forming synapses with unpulsed Raji cells was significantly lower than with SEE-pulsed Raji cells, and similar to MTOC PI of FMNL1-interfered cells (shFMNL1-YFP) forming synapses with SEE-pulsed Raji cells (*Figure 4—figure supplement 1B*). This confirms that antigenic stimulation at the IS indeed triggers MTOC polarization and validates our synapse model measurements. In addition, confirming our previous data (*Bello-Gamboa et al., 2020*), MTOC/MVB polarization was disrupted upon FMNL1 interference to the levels obtained with unpulsed Raji cells (*Figure 4A*, compare the first two rows, *Figure 4B*, *Figure 4—figure supplement 1B*), and was restored by YFP-FMNL1βWT expression (*Figure 4A*, third row and *Figure 4B*). Interestingly, non-phosphorylatable YFP-FMNL1βS1086A expression was unable to restore MVB/MTOC polarization to control levels (*Figure 4A*, fourth row and *Figure 4B*), whereas phosphomimetic YFP-FMNL1βS1086D expression restored MVB/MTOC polarization to the levels achieved by YFP-FMNL1βWT expression (*Figure 4A*, fifth row and *Figure 4B*). Comparable results were obtained by confocal microscopy (*Figure 4—figure supplement 2*). From confocal images, en face views of the IS interface were generated (*Figure 4—figure supplement 3*, *Figure 4—figure supplement 4*, and *Videos 1 and 2*). These images show the polarized MTOC or the polarized accumulation of MVB (measured as high MTOC and MVB PI values, in white types), respectively, in the cIS region of untransfected (control YFP⁻) C3 cells or cells expressing YFP-FMNL1WT or YFP-FMNL1S1086D, but not in cells transfected with shFMNL1-YFP or expressing YFP-FMNL1βS1086A (low MTOC and MVB PI values). Thus, phosphorylation of S1086 in FMNL1β appears to be essential for secretory polarized traffic toward the IS in T lymphocytes. Moreover, throughout the experiment, we observed that MTOC and MVB were polarizing together toward the IS. This fact, together with the previous work regarding segregation between MTOC movement and secretory granules traffic (*Chemin et al., 2012*; *Ma et al., 2007*; *Bertrand et al., 2013*; *Nath et al., 2016*), prompted us to simultaneously analyze both MTOC and MVB positioning with respect to the IS at the single cell level. Remarkably, we observed a robust linear correlation between MVB and MTOC PIs both in untransfected (control YFP⁻) C3 cells and cells transfected with shFMNL1-YFP (Pearson's linear correlation coefficients 0.96 and 0.95, respectively, *Figure 4—figure supplement 5*). Furthermore, it is noteworthy that MVB and MTOC centers of mass (MVB^C and MTOC^C, respectively) are very closely located in all the studied cell groups, regardless of polarization state (*Figure 4*, *Figure 4—figure supplement 2*). Although the MTOC and MVB did not efficiently polarize in cells transfected with shFMNL1-YFP, their MVB^C were still located next to MTOC^C (*Figure 4A*, second

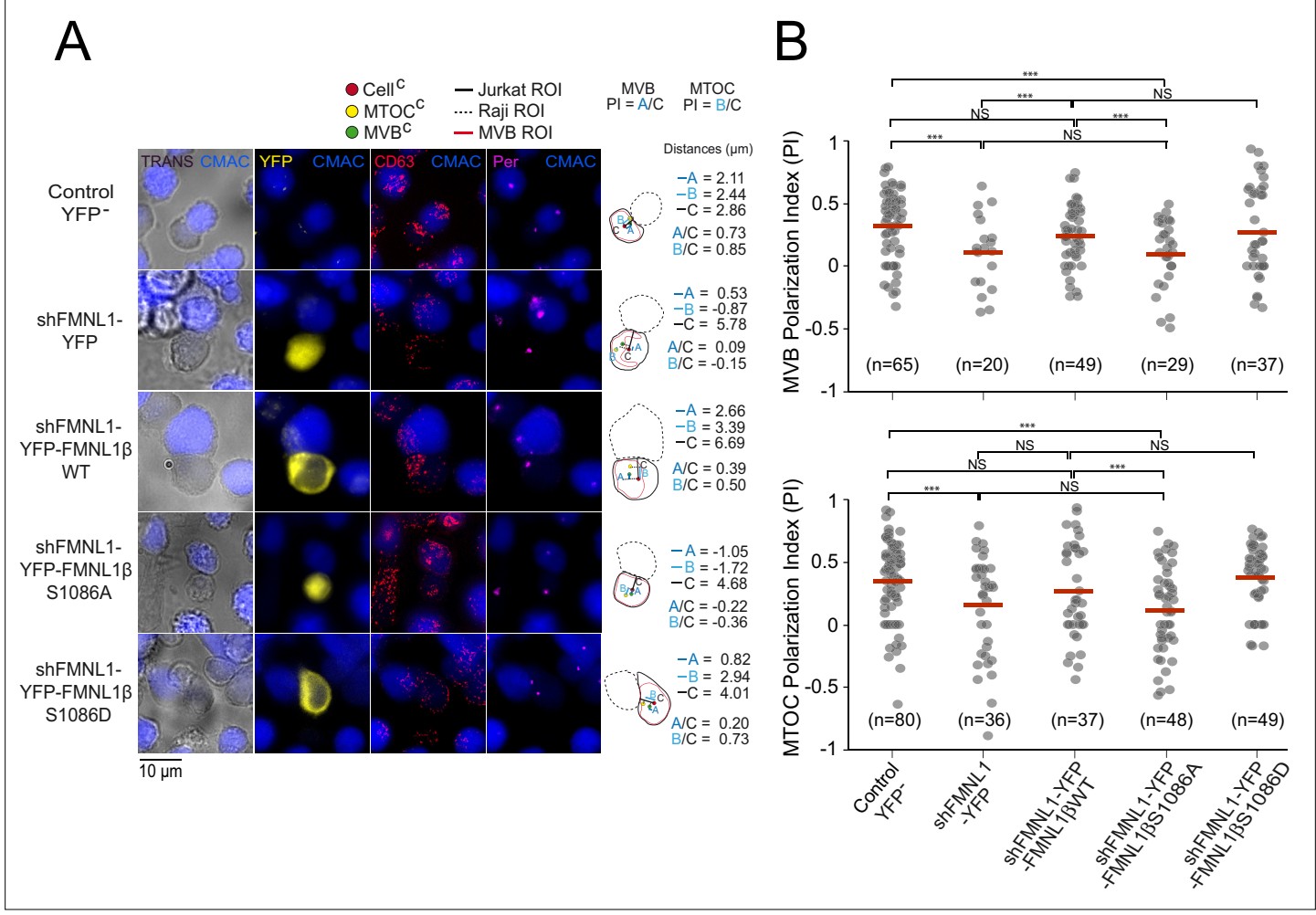

**Figure 4.** FMNL1β phosphorylation at S1086 is involved in microtubule-organizing center (MTOC)/multivesicular bodies (MVB) polarization toward the immune synapse (IS). C3 control clone was untransfected (Control YFP⁻) (first row) or transfected with FMNL1-interfering (shFMNL1-HA-YFP) (second row), or FMNL1-interfering expressing interference-resistant YFP-FMNL1βWT (shFMNL1-HA-YFP-FMNL1βWT) (third row), YFP-FMNL1βS1086A (fourth row), or YFP-FMNL1βS1086D (fifth row) constructs. Subsequently, cells were challenged with CMAC-labeled, SEE-pulsed Raji cells (blue) for 1 hr, fixed, stained with anti-pericentrin (magenta) to label the MTOC and anti-CD63 (red) to label MVB, and imaged by epifluorescence microscopy. (**A**) Representative maximum intensity projection (MIP) images with the indicated merged channels for each of the specified cell groups, along with a schematic diagram on the right representing the measured parameters used to calculate the MTOC and MVB polarization index (PI). This includes the distance in microns between the MTOC[C] (or MVB[C]) projection on the vector defined by the Cell[C]-synapse axis and the Cell[C] ('B' or 'A' distance, respectively), and the distance between the Cell[C] and the synapse ('C' distance). (**B**) Dot plots of MVB and MTOC PI from each of the indicated cell groups, corresponding to the indicated number of synapses from a experiment similar to that described in (**A**) are depicted. NS, not significant. ***, p<0.05. Results and ANOVA are representative of data from several independent experiments (n=3) with similar results.

The online version of this article includes the following figure supplement(s) for figure 4:

**Figure supplement 1.** SEE-induced polarization of multivesicular bodies (MVB) and microtubule-organizing center (MTOC) to the immune synapse (IS) and calculation of their polarization indexes.

**Figure supplement 2.** FMNL1β phosphorylation at S1086 is involved in microtubule-organizing center (MTOC)/multivesicular bodies (MVB) polarization toward the immune synapse (IS).

**Figure supplement 3.** T cell en face analysis of F-actin and microtubule-organizing center (MTOC) distribution at the immune synapse (IS) interface.

**Figure supplement 4.** T cell en face analysis of F-actin and multivesicular bodies (MVB) distribution at the immune synapse (IS) interface.

**Figure supplement 5.** Correlation between microtubule-organizing center (MTOC) and multivesicular bodies (MVB) polarization indexes.

**Video 1.** T cell en face visualization of F-actin and microtubule-organizing center (MTOC) at the immune synapse (IS) interface. Untransfected C3 cells (Control YFP⁻) (first column) or C3 cells transfected with either the bi-cistronic, YFP expression plasmid interfering all FMNL1 isoforms (shFMNL1-HA-YFP) (second column), or FMNL1 interfering and expressing interference-resistant YFP-FMNL1βWT (shFMNL1-HA-YFP-FMNL1βWT) (third column), YFP-FMNL1βS1086A (fourth column), or YFP-FMNL1βS1086D (fifth column) constructs, were mixed with CMAC-labeled, SEE-pulsed Raji cells (blue) attached to slides as described in Materials and methods. After 1 hr of culture, synaptic conjugates were fixed and labeled for F-actin (acquired in magenta, changed to blue) and for γ-tubulin (red). Fixed whole cells (top row) and synapse contact areas (bottom row) were imaged with confocal microscopy and Z stacks were processed to generate en face, zx views of the synaptic interface, as indicated in Materials and methods. The video (10 fps) shows the generation via a 90° turn of the en face view of the Jurkat cell where the MTOC is displayed at the central region of the immune synapse (cIS) in the cell groups that underwent polarization. This video is related to *Figure 4—figure supplement 3*.
https://elifesciences.org/articles/96942/figures#video1

**Video 2.** T cell en face visualization of F-actin and multivesicular bodies (MVB) at the immune synapse (IS) interface. Untransfected C3 cells (Control YFP⁻) (first column) or C3 cells transfected with either the bi-cistronic, YFP expression plasmid interfering all FMNL1 isoforms (shFMNL1-HA-YFP) (second column), or FMNL1 interfering and expressing interference-resistant YFP-FMNL1βWT (shFMNL1-HA-YFP-FMNL1βWT) (third column), YFP-FMNL1βS1086A (fourth column), or YFP-FMNL1βS1086D (fifth column) constructs, were mixed with CMAC-labeled, SEE-pulsed Raji cells (blue) attached to slides as described in Materials and methods. After 1 hr of culture, synaptic conjugates were fixed and labeled for F-actin (acquired in magenta, changed to blue) and for CD63 (red). Fixed whole cells (top row) and synapse contact areas (bottom row) were imaged with confocal microscopy and Z stacks were processed to generate en face, zx views of the synaptic interface, as indicated in Materials and methods. The video (10 fps) shows the generation via a 90° turn of the en face view of the Jurkat cell where the MVB are accumulated at the central region of the immune synapse (cIS) if they are polarized. When MVB are not polarized, some MVB can still be observed at cIS because they are scattered throughout the cell. This video is related to *Figure 4—figure supplement 4*.
https://elifesciences.org/articles/96942/figures#video2

row), and equivalent results were obtained in all the analyzed cell groups (*Figure 4*). Thus, phosphorylation of S1086 in FMNL1β appears to be essential for both MTOC and MVB polarization toward the IS in T lymphocytes.

## YFP-FMNL1βS1086D phosphomimetic mutant expression does not rescue MTOC polarization in PKCδ-interfered T lymphocytes

The former results supported the contribution of FMNL1β and its phosphorylation at S1086 to MVB/MTOC polarization, but did not address the sufficiency in this process of PKCδ-controlled, S1086 FMNL1β phosphorylation. In this context, we have previously shown that FMNL1β lies downstream of PKCδ in the same pathway controlling MTOC/MVB polarization (*Bello-Gamboa et al., 2020*; *Calvo and Izquierdo, 2021*). Thus, in order to analyze this important issue we transfected the PKCδ-interfered P5 and P6 clones with the different plasmids and compared the MTOC PI among the different cell groups, as previously done in C3 clone. We have previously shown that neither PKCδ nor FMNL1 interference affects IS conjugate formation, since P5 and P6 clones and FMNL1-interfered C3 clone formed IS with SEE-pulsed Raji cells (*Herranz et al., 2019*; *Bello-Gamboa et al., 2020*). In addition, IS formed by P5 clone with SEE-pulsed Raji cells (fifth group in *Figure 4—figure supplement 1B*) had a MTOC PI comparable to those of IS formed by C3 clone with unpulsed Raji cells or P5 clone with unpulsed Raji cells (first and fourth group, respectively, in *Figure 4—figure supplement 1B*), supporting that both FMNL1 and PKCδ interference fully inhibited MTOC polarization.

In contrast to what was found in the control C3 clone (*Figure 4* and *Figure 4—figure supplement 2*) or C9 clone (not shown), YFP-FMNL1βWT expression in PKCδ and FMNL1-interfered P5 clone (*Figure 5*) or P6 clone (not shown) did not restore MTOC PI to the values observed in control YFP⁻ C3 cells, which is compatible with the idea that FMNL1β lies downstream of PKCδ and its phosphorylation is indispensable for MTOC polarization. Consistently, expression in P5 clone of YFP-FMNL1βS1086A,

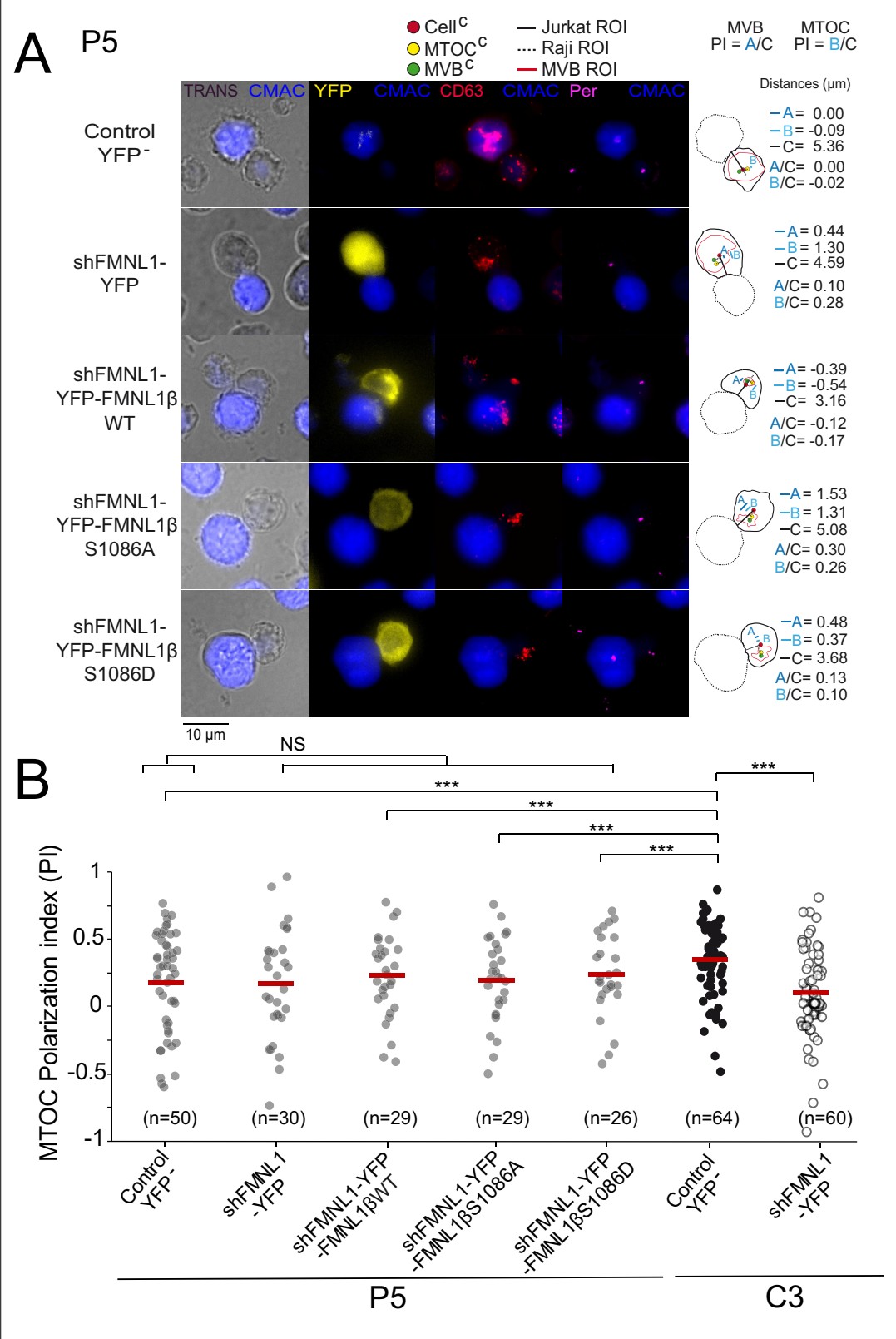

**Figure 5.** YFP-FMNL1βS1086D expression does not rescue deficient microtubule-organizing center (MTOC) polarization in PKCδ-interfered cells. PKCδ-interfered P5 clone was untransfected (Control YFP⁻) (first row) or transfected with FMNL1-interfering (shFMNL1-HA-YFP) (second row), FMNL1-interfering expressing interference-resistant YFP-FMNL1βWT(shFMNL1-HA-YFP-FMNL1βWT) (third row), YFP-FMNL1βS1086A (fourth row), or YFP-

*Figure 5 continued on next page*

*Figure 5 continued*

FMNL1βS1086D (fifth row) constructs. In parallel, C3 control clone was untransfected (Control YFP⁻) or transfected with FMNL1-interfering (shFMNL1-HA-YFP) construct. Subsequently, cells were challenged with CMAC-labeled, SEE-pulsed Raji cells (blue) for 1 hr, fixed, stained with anti-pericentrin (magenta) to label the MTOC and anti-CD63 (red) to label multivesicular bodies (MVB), and imaged by epifluorescence microscopy. (**A**) Representative maximum intensity projection (MIP) images with the indicated merged channels for each of the specified cell groups along with their corresponding diagrams as in *Figure 4*, representing the measured parameters used for calculating the MTOC and MVB polarization index (PI) are shown. This includes the distance in microns between the MTOC$^C$ (or MVB$^C$) projection on the vector defined by the Cell$^C$-synapse axis and the Cell$^C$ ('B' or 'A' distance, respectively), and the distance between the Cell$^C$ and the synapse ('C' distance). (**B**) Dot plots of MTOC PI of each cell group. Untransfected (Control YFP⁻) and FMNL1-interfering (shFMNL1-HA-YFP) C3 clone synapses corresponding to the data shown in *Figure 4—figure supplement 2B* were included in the far-right columns, as a reference. NS, not significant; ***, p<0.05. Results and ANOVA are representative of data from several independent experiments (n=3) with similar results.

a variant non-phosphorylatable by PKC, was also unable to enhance MTOC PI to the values observed in control YFP⁻ C3 cells (*Figure 5B*). Moreover, phosphomimetic YFP-FMNL1βS1086D expression, which restored MVB/MTOC PI to the values achieved by YFP-FMNL1βWT expression in the C3 clone (*Figure 4* and *Figure 4—figure supplement 2*), did not restore MTOC PI in PKCδ and FMNL1-interfered P5 cells to the levels observed in control YFP⁻ C3 cells (*Figure 5B*). Similarly, FMNL1βS1086D expression was unable to revert the deficient MTOC polarization occurring in P6 (not shown), a different PKCδ-interfered clone previously described (*Herranz et al., 2019*). We have shown that the deficient MTOC/MVB polarization in the PKCδ-interfered P5 and P6 clones is due exclusively to the reduction in PKCδ expression, since transient re-expression of PKCδ in all these clones recovered MTOC/MVB polarization to control levels (*Herranz et al., 2019*). Since both the FMNL1 interference and FMNL1 variant re-expression for IS experiments were performed in transient assays (2–4 days after transfection), there was no chance for any clonal variation in these short-time experiments. Taken together, these results show that although FMNL1β phosphorylation at S1086 is necessary, it does not seem to be sufficient for MTOC polarization, at least in cells lacking PKCδ.

## FMNL1β translocation to the IS is independent of S1086 phosphorylation and PKCδ

Previous results have shown that FMNL1, apart from being mainly located at the cytosol and centrosomal areas, was also found located at the IS in a small percentage of synaptic conjugates, in endpoint IS experiments (*Gomez et al., 2007*; *Bello-Gamboa et al., 2020*). If confirmed, the possibility of an IS-induced FMNL1 translocation to the IS may provide the molecular basis underlying a potential regulatory effect of FMNL1 on cortical actin cytoskeleton reorganization at the IS as well as on MTOC/MVB and secretion granule polarization, occurring both in CTL and Th cells, as suggested by several authors (*Ritter et al., 2013*; *Ritter et al., 2015*; *Chemin et al., 2012*; *Ueda et al., 2015*; *Calvo and Izquierdo, 2021*). However, although it could be inferred (*Gomez et al., 2007*; *Murugesan et al., 2016*) that the active translocation of any FMNL1 isoform from cytosol toward the IS is induced upon IS formation, it has not been formally demonstrated yet (*Gomez et al., 2007*; *Murugesan et al., 2016*). This is probably due to the fact that most early studies have used an end-point approach that does not allow to analyze the incipient IS (*Gomez et al., 2007*; *Kupfer and Singer, 1989*; *Calvo and Izquierdo, 2018*; *Bello-Gamboa et al., 2019*). Since FMNL1β is responsible for MTOC/MVB polarization to the IS (*Bello-Gamboa et al., 2019*), we analyzed FMNL1β subcellular location in developing synapses by time-lapse, epifluorescence microscopy (*Calvo and Izquierdo, 2018*; *Bello-Gamboa et al., 2019*). For this analysis, FMNL1-interfered, YFP-FMNL1βWT-expressing C3 clone was challenged with SEE-pulsed Raji cells. As evidenced in the example shown in *Figure 6—video 1* (upper panel) and *Figure 6A* (upper row), a transient relocation of initially cytosolic YFP-FMNL1βWT to the IS (white arrow) was observed in these cells. The average (± SD) duration of YFP-FMNL1βWT accumulation at the IS in C3 clone was around 6 min (6 min 18 s±1 min 34 s, n=6 synapses).

Considering the major role of PKCα in FMNL2 relocation from the plasma membrane to the endosomal membrane (*Wang et al., 2015*), the high similarity between FMNL2 and FMNL1β C-terminal regions (*Figure 1*) and the reported PKCδ-mediated FMNL1β phosphorylation (*Bello-Gamboa et al., 2020*), we next aimed to analyze the role of PKCδ in YFP-FMNL1βWT translocation to the IS. To this

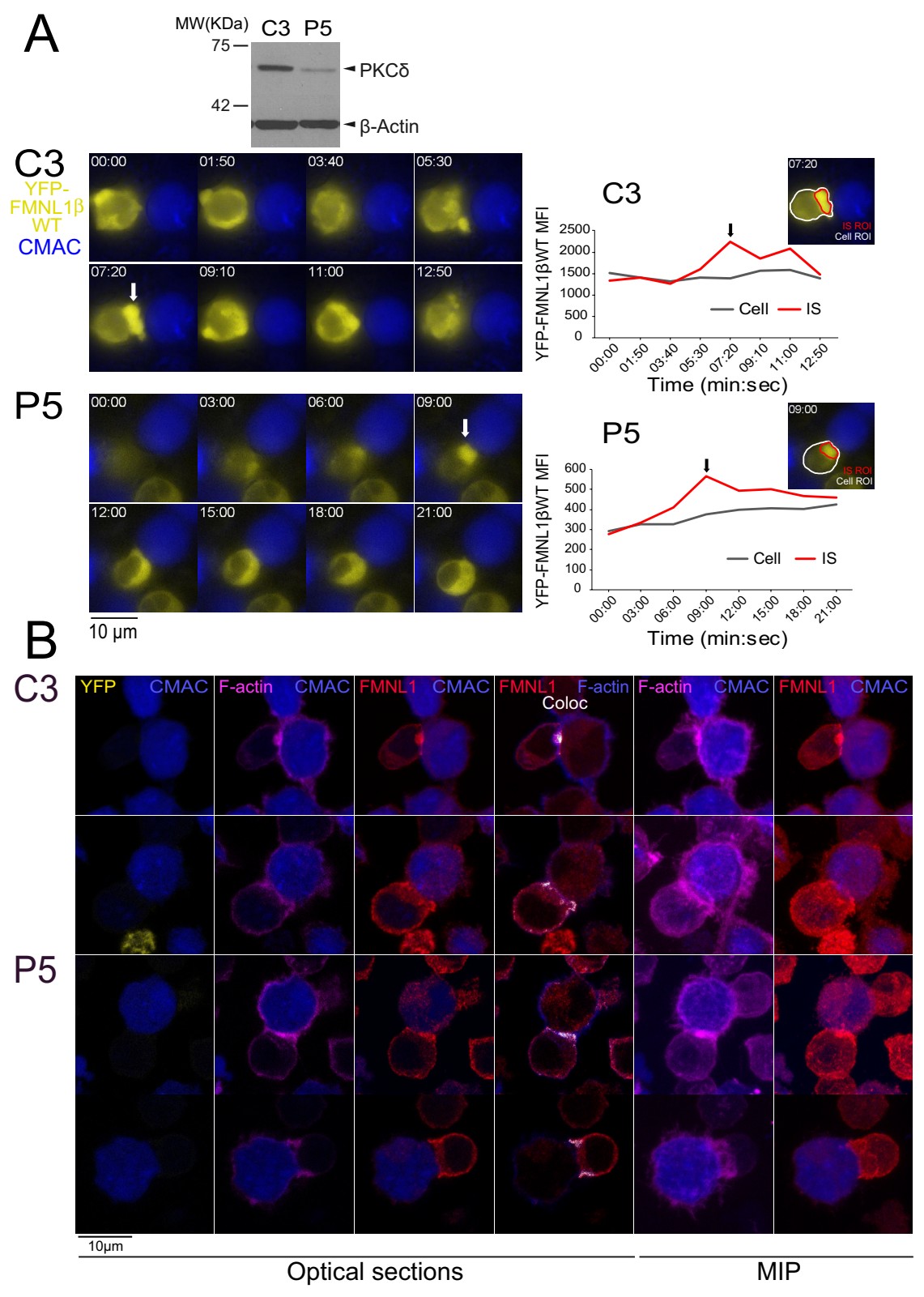

**Figure 6.** FMNL1β recruitment to the immune synapse (IS) is protein kinase C δ (PKCδ)-independent. C3 control and P5 PKCδ-interfered cells were transfected with FMNL1-interfering expressing interference-resistant YFP-FMNL1βWT (shFMNL1-HA-YFP-FMNL1βWT) plasmid. Subsequently, both transfected clones were simultaneously challenged with CMAC-labeled, SEE-pulsed Raji cells (blue) attached to slides and time-lapse acquisition of emerging synapses was performed as indicated in Materials and methods. The videos (7 fps) (**Figure 6—video 1**) were captured and in (**A**), left,

*Figure 6 continued on next page*

*Figure 6 continued*

representative frames from videos of each clone are shown. White arrows indicate accumulations of YFP-FMNL1βWT at the IS. Western blot (WB) analysis of cell lysates from both clones (top inset) shows PKCδ silencing in P5 clone. In the right side, YFP-FMNL1βWT mean fluorescence intensity (MFI) within the cell regions of interest (ROI) (gray line) and the IS ROI (red line) are represented. The inserts in the graphs include the cell ROI (white) and the IS ROI (red) used for the time-lapse measurements on representative frames for both clones. Results are representative of the data from several videos (n=6 for each clone) with similar results. (**B**) Control C3 (upper rows) and P5 PKCδ-interfered cells (lower rows) were simultaneously challenged with CMAC-labeled SEE-pulsed Raji cells (blue) attached to slides. After 1 hr, synapses were fixed and immunofluorescence developed with anti-FMNL1 (red) to label endogenous FMNL1 and phalloidin (magenta) to label F-actin. Synapses were imaged with confocal fluorescence microscopy and colocalization pixels between FMNL1 (red) and F-actin (acquired in magenta, changed to blue in the fourth column) are represented in white. Representative optical sections of synapses formed by both clones are shown in the left columns. The colocation coefficients were: first row, Pearson = 0.50, Manders-=0.877; second row, Pearson = 0.431, Manders = 0.864; third row: Pearson = 0.434, Manders = 0.838; fourth row, Pearson = 0. 484, Manders = 0.794. Maximum intensity projection (MIP) images of the same synapses are shown in the two far-right columns. Results and ANOVA are representative of data from several independent experiments (n=3) with similar results.

The online version of this article includes the following video, source data, and figure supplement(s) for figure 6:

**Source data 1.** Original, uncropped WBs corresponding to the upper panel in *Figure 6A* in PDF format.

**Source data 2.** Original, uncropped WBs corresponding to the upper panel in *Figure 6A* in TIF format.

**Figure supplement 1.** FMNL1 is located at the immune synapse (IS) developed by primary T lymphoblasts and dual chimeric antigen receptor (CAR) T cells recognizing CD19/CD22.

**Figure supplement 2.** YFP-FMNL1βS1086A and S1086D mutants are recruited to the immune synapse (IS).

**Figure supplement 3.** FMNL1 and FMNL1β accumulation at the immune synapse (IS) revealed by epifluorescence microscopy.

**Figure supplement 4.** YFP-FMNL1β variants accumulation at the immune synapse (IS) revealed by confocal fluorescence microscopy.

**Figure 6—video 1.** YFP-FMNL1βWT is recruited to the immune synapse (IS) independently of protein kinase C δ (PKCδ).

https://elifesciences.org/articles/96942/figures#fig6video1

end we analyzed YFP-FMNL1βWT translocation upon IS formation in the PKCδ-interfered P5 clone (*Herranz et al., 2019*; *Figure 6—video 1* and *Figure 6*). The results in P5 clone showed a clear translocation of YFP-FMNL1WT to the IS (white arrow), with an accumulation of around 8 min at the IS (8 min 55 s±3 min 2 s, n=6 synapses, no significant differences with the C3 clone), so YFP-FMNL1βWT translocation to the IS in P5 and C3 clones showed comparable kinetics.

Moreover, to directly analyze the specific involvement of S1086 phosphorylation in this process, we challenged FMNL1-interfered, YFP-FMNL1βS1086A and YFP-FMNL1βS1086D-expressing C3 cells with SEE-pulsed Raji cells. As shown in *Video 3* and *Figure 6—figure supplement 2*, both YFP-FMNL1βS1086A and YFP-FMNL1βS1086D transiently relocated to the IS. The average time of YFP-FMNL1βS1086A accumulation at the IS was around 7 min (7 min 30 s+1 min 58 s, n=6 synapses), whereas YFP-FMNL1βS1086D showed an average accumulation time of around 6 min (6 min 52 s+2 min 9 s, n=7 synapses; no significant differences between YFP-FMNL1βS1086A and YFP-FMNL1βS1086D, nor between any of the mutants and YFP-FMNL1βWT). Therefore, these results confirm that the translocation and accumulation of FMNL1β in the IS region of Jurkat cells is a process that occurs independently of both PKCδ and S1086 FMNL1β phosphorylation.

A more detailed study of subsynaptic location of endogenous FMNL1 in C3 control clone using fixed synapses demonstrated that FMNL1 accumulated at the cIS but also at the edges of the synapse, around the dSMAC, both in C3 control clone (*Figure 6B*, first and second rows, respectively) and in P5 PKCδ-interfered clone (*Figure 6B*, third and fourth rows). At these subsynaptic locations, FMNL1 colocalized with F-actin (white pixels), which is compatible with the idea that FMNL1 regulates F-actin at the IS. To extend these results obtained in Jurkat cells, we have performed experiments using primary T cells developing synapses mediated by TCR recognizing SEE plus SEB superantigens and a dual chimeric antigen receptor (CAR) recognizing CD19/CD22 on Raji cells (*Figure 6—figure supplement 1A and B*, respectively). As shown in this figure, endogenous FMNL1 colocalizing with F-actin at the IS was also found in both types of synapses developed by primary T lymphocytes.

Subsequently, we studied the subsynaptic locations of YFP-FMNL1βWT, YFP-FMNL1βS1086A, and YFP-FMNL1βS1086D in C3 control clone, to identify whether S1086 phosphorylation affected these locations. The results from fixed synapses, obtained through both epifluorescence (*Figure 6—figure supplements 2 and 3*) and confocal microscopy (*Figure 6—figure supplement 4*), revealed the presence of accumulations of the three YFP-FMNL1β variants in the IS. Additionally, in *Figure 6—figure*

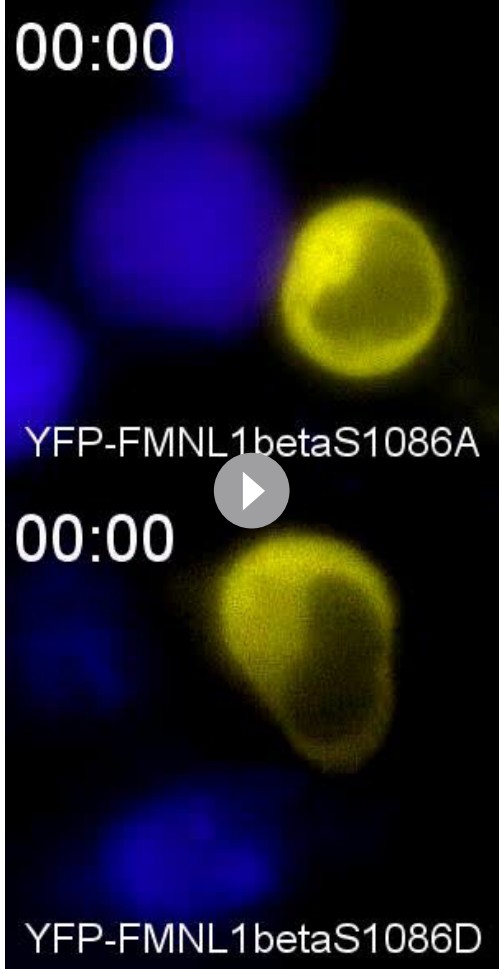

**Video 3.** YFP-FMNL1βS1086A and S1086D mutants are recruited to the immune synapse (IS). YFP-FMNL1βS1086A and S1086D-expressing, C3 control clone cells (upper panel and lower panel, respectively) were mixed with CMAC-labeled, SEE-pulsed Raji cells (blue) attached to slides as described in Materials and methods. Subsequently, the conjugates were imaged by time-lapse microscopy (7 fps) for the indicated times. In both panels, CMAC (blue) and YFP-FMNL1β (yellow) channels were merged. This video is related to *Figure 6—figure supplement 2*.

https://elifesciences.org/articles/96942/figures#video3

*supplement 4*, it can be observed that the three YFP-FMNL1β variants colocalized with F-actin at the IS.

Taken together, these observations show that, upon IS formation, FMNL1β translocates to the IS, with both PKCδ and S1086 phosphorylation being dispensable for the translocation to the IS of both endogenous FMNL1 and YFP-FMNL1β variants.

## S1086 phosphorylation of FMNL1β is necessary for cortical F-actin reorganization during MTOC/MVB polarization

The observations that YFP-FMNL1β translocated to the IS, that cortical F-actin remodeling was considered to be sufficient for centrosome polarization in CTL (*Ritter et al., 2015*), and that cortical F-actin remodeling was associated with MTOC and secretory granule polarization in CD4[+] cells (*Chemin et al., 2012*; *Le Floc'h and Huse, 2015*), prompted us to study a potential role of FMNL1β and its phosphorylation at S1086 in synaptic F-actin architecture. To address this point, fixed synapses formed by the C3 clone transfected with the previously described plasmids were stained with phalloidin AF647 to label F-actin and imaged by confocal fluorescence microscopy. The relative area of the F-actin-low region at the cIS (Fact-low cIS area/IS area) in the IS interface, which mirrors cortical F-actin reorganization at the IS, was measured as indicated in Materials and methods and represented for all cell groups. As seen in a representative example in *Figure 7A*, the dot plot in *Figure 7B* and *Figure 7—video 1*, the F-actin-low region at the cIS was smaller in FMNL1-interfered cells than in untransfected control YFP[-] cells (*Figure 7A*, second and first columns, respectively), as it was also in PKCδ-interfered cells (*Herranz et al., 2019*; *Bello-Gamboa et al., 2020*). Of note, YFP-FMNL1βWT expression restored F-actin depletion at the cIS to control levels (*Figure 7B*). In contrast, YFP-FMNL1βS1086A expression did not rescue F-actin depletion to control levels, whereas YFP-FMNL1βS1086D did (*Figure 7B*). Thus, FMNL1β phosphorylation at S1086 appears to be necessary for F-actin reorganization at the IS, as it was for MTOC/MVB polarization (*Figure 4*).

### Three-dimensional FMNL1β distribution at the synapse

We observed a positive effect of FMNL1β phosphorylation at S1086 on cortical F-actin rearrangement at the IS (former paragraph and *Figure 7*) and YFP-FMNL1β translocated to the cIS, but also to the dSMAC (*Figure 6—video 1*, *Video 3*, *Figure 6*, and *Figure 6—figure supplements 2 and 3*). To investigate this further, we analyzed in more detail the subsynaptic distribution of the endogenous FMNL1, YFP-FMNL1βWT, and the YFP-FMNL1βS1086A and S1086D mutants with respect to the actin cytoskeleton. To this end, fixed synapses formed by the C3 clone transfected with the

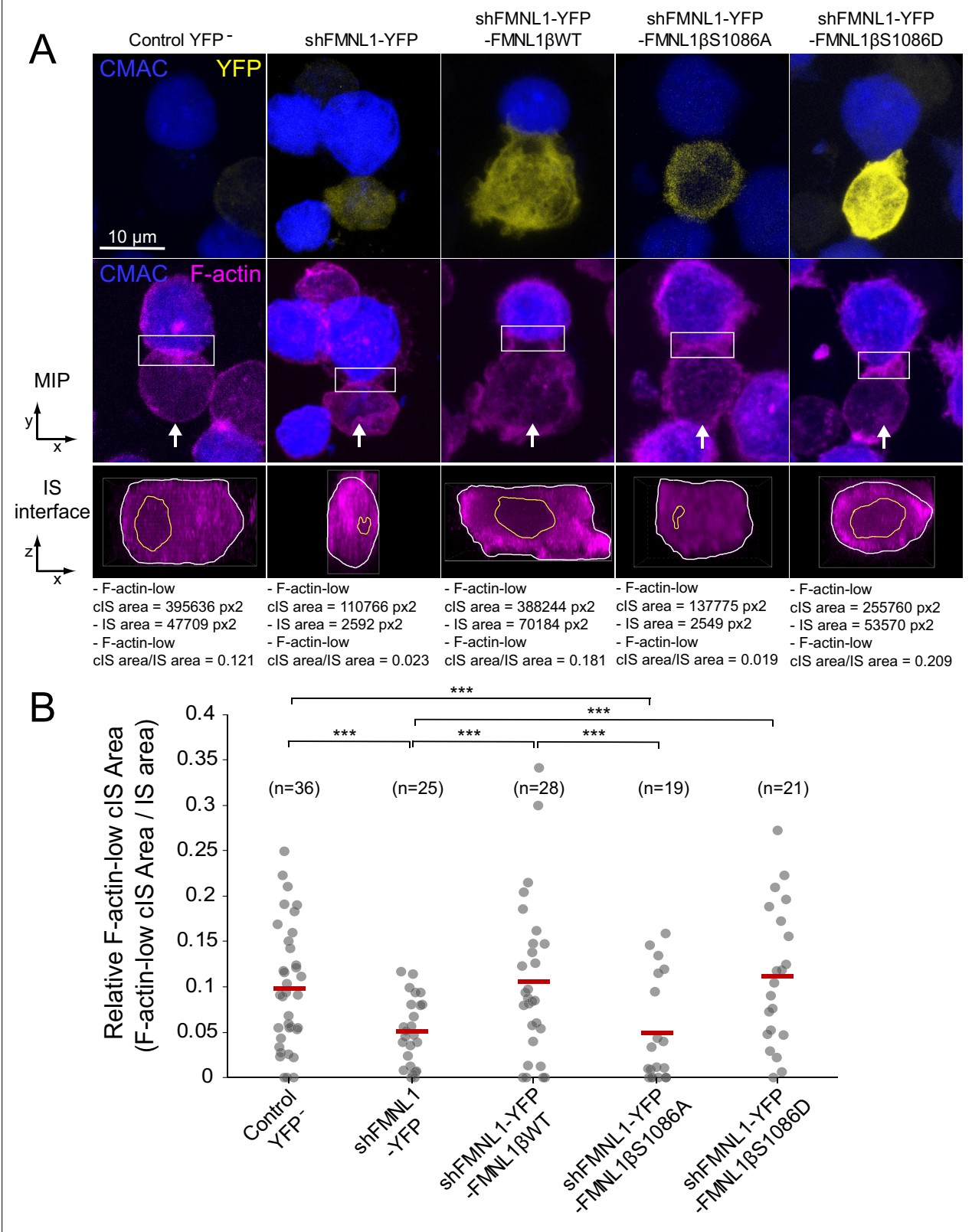

**Figure 7.** FMNL1β phosphorylation at S1086 regulates the F-actin architecture of the immune synapse (IS). C3 control clone was untransfected (Control YFP⁻) (first column) or transfected with FMNL1-interfering (shFMNL1-HA-YFP) (second column), FMNL1-interfering expressing interference-resistant YFP-FMNL1βWT (shFMNL1-HA-YFP-FMNL1βWT) (third column), YFP-FMNL1βS1086A (fourth column), or YFP-FMNL1βS1086D (fifth column) constructs. Subsequently, cells were challenged with CMAC-labeled, SEE-pulsed Raji cells (blue) for 1 hr, fixed, stained with phalloidin AF647 (magenta) to label

*Figure 7 continued on next page*

Figure 7 continued

F-actin, and imaged by confocal microscopy. (**A**) The upper rows display the top, yx views corresponding to the maximum intensity projection (MIP) images of the specified, merged channels of a representative example from each of the indicated cell groups. White arrows indicate the direction to visualize the en face views of the IS (IS interface) enclosed by the regions of interest (ROIs) (white rectangles), as shown in *Figure 7—video 1*. In the lower panels, the enlarged ROIs (2× zoom) used to generate the IS interface, zx images of each cell group are shown. The areas of the F-actin-low region at the central region of the IS (cIS) (Fact-low cIS area) (yellow line) and the synapse (IS area) (white line) were defined and measured as indicated in Materials and methods, and the relative area of the F-actin-low region at the cIS (Fact-low cIS area/IS area) was calculated and represented. (**B**) Relative area (Fact-low cIS area/IS area) dot plot distributions and average area ratios (red horizontal lines) for the indicated number of IS conjugates developed by each cell group are shown. This figure is related to *Figure 7—video 1*. NS, not significant; ***, p≤0.05. Results are representative of data from several independent experiments (n=3) with similar results.

The online version of this article includes the following video for figure 7:

**Figure 7—video 1.** S1086 phosphorylation in FMNL1β regulates the F-actin architecture of the immune synapse (IS).

https://elifesciences.org/articles/96942/figures#fig7video1

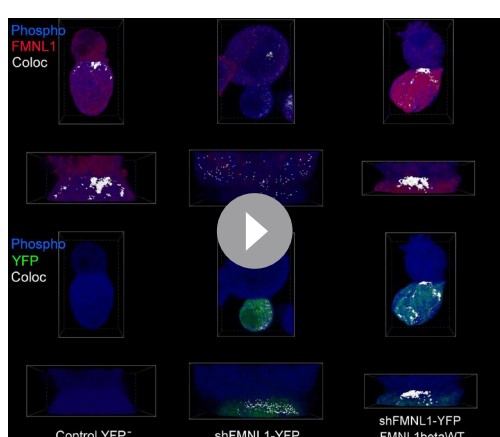

**Video 4.** Colocalization of FMNL1 and anti-phospho-Ser PKC substrate at the immune synapse (IS) interface. Untransfected C3 cells (Control YFP⁻) (first left column) or C3 cells transfected with either the bi-cistronic, YFP expression plasmid interfering all FMNL1 isoforms (shFMNL1-HA-YFP) (second column), or FMNL1 interfering and re-expressing interference-resistant YFP-FMNL1βWT (shFMNL1-HA-YFP-FMNL1βWT) (third column) constructs were mixed with CMAC-labeled, SEE-pulsed Raji cells (blue) attached to slides, as described in Materials and methods. After 1 hr of culture, synaptic conjugates were fixed and labeled with anti-FMNL1 (red) and anti-Phospho-Ser PKC substrate (magenta, changed to blue). YFP fluorescence was in yellow (changed to green). Upper panels, the video (10 fps) shows fixed whole cells (top row) and synapse contact areas (second row) that were imaged with confocal microscopy and Z stacks were processed to generate en face, zx views of the synaptic interface (final frame in the video), as indicated in Materials and methods. Colocalization pixels (white) of FMNL1 (red) with anti-phospho-Ser PKC substrate (blue) are shown. Lower panels, same as upper panels, but the top views and the IS interfaces were generated and stained with anti-phospho-Ser PKC substrate (blue) merged to YFP fluorescence (green). This video is related to *Figure 8—figure supplement 1*.

https://elifesciences.org/articles/96942/figures#video4

different plasmids were stained with phalloidin, anti-FMNL1, and anti-phospho-Ser PKC substrate antibody and imaged by confocal fluorescence microscopy (*Figure 8—video 1*, *Video 4*, *Figure 8*, and *Figure 8—figure supplement 1*). *Figure 8A and B* shows representative examples of synaptic conjugates yx views (first row) and cropped synapse areas (white rectangles in first row) used to generate zx IS interfaces (second row), as shown in *Figure 8—video 1*, *Video 4*. Subsequently, the colocalization pixels (white) along all the Z stack at the IS interface (interface colocalization) were determined as indicated in Materials and methods. This involved merging the indicated fluorescence channels (*Figure 8A*, second row, F-actin in blue and anti-FMNL1 in red; *Figure 8B*, second row, F-actin in blue and YFP fluorescence in green) on the IS interfaces of the synaptic areas generated as shown in *Figure 8—video 1*. MFI profile of each channel (F-actin in blue, anti-FMNL1 in red, YFP in green, and colocalization in gray) along the indicated white arrows are shown below each IS interfaces. As shown in *Figure 8*, and consistently with results from *Figure 7*, both control YFP⁻ (first column) and YFP-FMNL1βWT-expressing cells (third and fourth columns) exhibited a wide F-actin depletion area at the cIS and an F-actin accumulation at the dSMAC. Interestingly, in some synapses from control YFP⁻ (first column) and YFP-FMNL1βWT-expressing cells (fourth column), F-actin and FMNL1 colocalized (white pixels) at certain small areas at the wide F-actin-low region at the cIS, although the MFI profiles show relatively less F-actin and FMNL1 at the cIS than at the dSMAC. In other synapses from both control YFP⁻ (not shown) and YFP-FMNL1βWT-expressing cells (third column), F-actin and FMNL1 colocalization occurred at the F-actin-rich area corresponding to the dSMAC (*Figure 8A*), which was consistent with the data from *Figure 6B*. In contrast, and

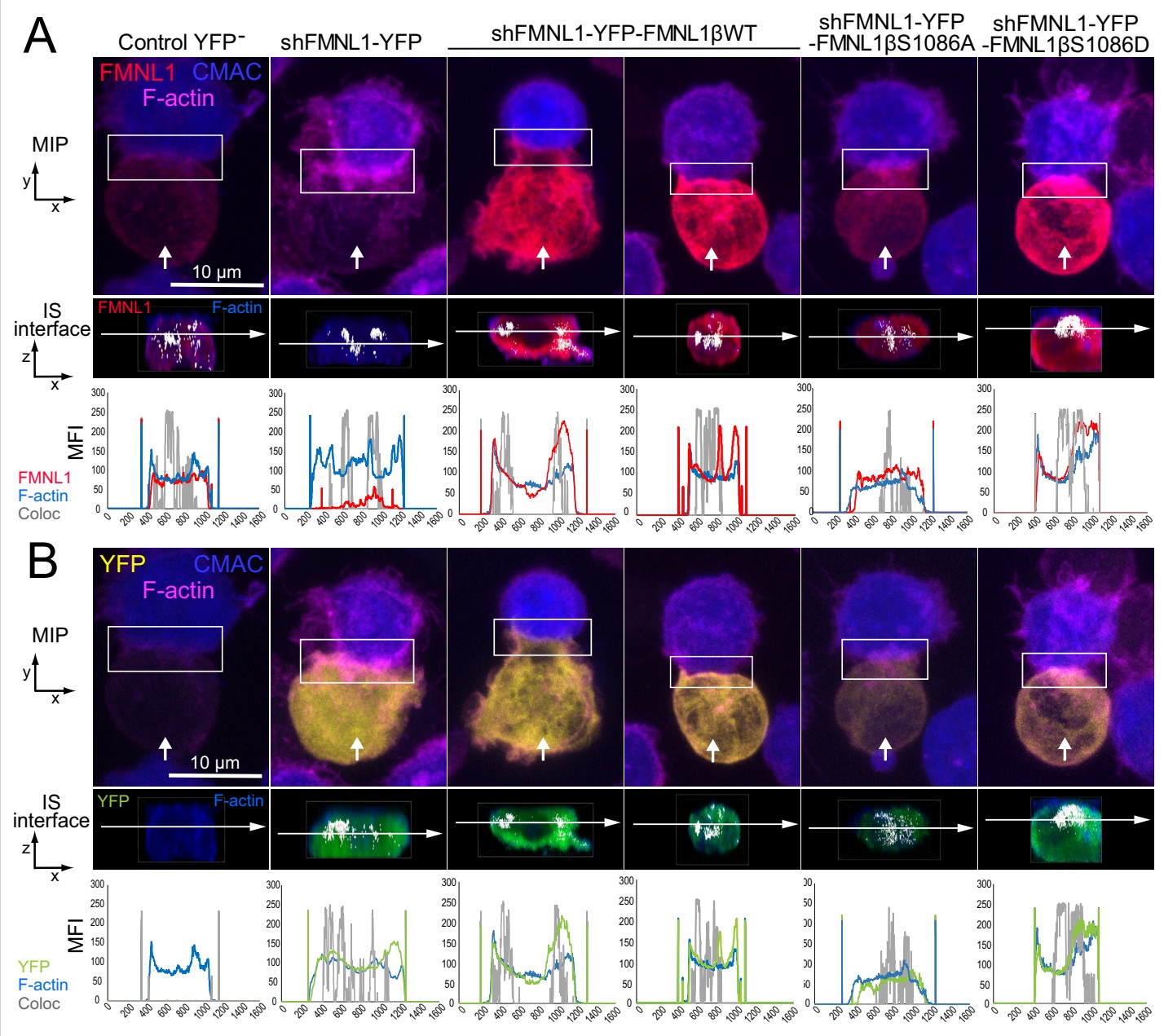

**Figure 8.** Three-dimensional distribution and colocalization of FMNL1 and F-actin at the immune synapse (IS) interface. C3 control clone was untransfected (Control YFP⁻) (first column) or transfected with FMNL1-interfering (shFMNL1-HA-YFP) (second column), FMNL1-interfering expressing interference-resistant YFP-FMNL1βWT(shFMNL1-HA-YFP-FMNL1βWT) (third and fourth column), YFP-FMNL1βS1086A (fifth column) or YFP-FMNL1βS1086D (sixth column) constructs. Subsequently, cells were challenged with CMAC-labeled SEE-pulsed Raji cells (blue) for 1 hr, fixed, and stained with phalloidin (magenta) and anti-FMNL1 (red). The corresponding shFMNL1 construction is in yellow, and synaptic conjugates were imaged by confocal fluorescence microscopy. Please realize that for the IS interface and since interface colocalization in NIS-AR only works with red, green, and blue channels, F-actin (acquired in magenta) was changed to blue color, and YFP (acquired in yellow) was changed to green in the second row of each panel. (**A**) The upper row includes top, yx views corresponding to the maximum intensity projection (MIP) images of representative examples of each cell group. Vertical white arrows indicate the direction to visualize the en face views of the IS (IS interface) enclosed by the regions of interest (ROIs) (white rectangles), as shown in *Figure 8—video 1*. In the second row, the enlarged ROIs (2× zoom) used to generate the IS interface, zx images of each cell group are shown. Subsequently, interface colocalization pixels (white) were generated by merging the indicated channels in the second row of each panel (F-actin in blue merged to anti-FMNL1 in red), at the IS interfaces of the synaptic areas (generated as shown in *Figure 8—video 1*). The last frame of these videos corresponds to the en face view (interface) (second row in both panels). Mean fluorescence intensity (MFI) profiles along the indicated line (horizontal white arrow) of each separate channel (F-actin in blue, anti-FMNL1 in red) and the colocalization pixels (gray) are shown below the IS interfaces. (**B**) Same as (**A**), but the top views show YFP-expressing constructs signal instead of the anti-FMNL1 signal. The IS interfaces and the MFI

*Figure 8 continued on next page*

*Figure 8 continued*

profiles show F-actin (magenta changed to blue) and YFP (yellow changed to green). This figure is related to *Figure 8—video 1*. At least six synapses of each cell group were analyzed. Results are representative of data from several independent experiments (n=3) with similar results.

The online version of this article includes the following video and figure supplement(s) for figure 8:

**Figure supplement 1.** Colocalization of FMNL1 and anti-phospho-Ser PKC substrate at the immune synapse (IS) interface.

**Figure supplement 2.** STED colocalization of FMNL1 and anti-phospho-Ser protein kinase C (PKC) substrate at the immune synapse (IS).

**Figure 8—video 1.** Colocalization of FMNL1 with F-actin at the immune synapse (IS) interface.

https://elifesciences.org/articles/96942/figures#fig8video1

consistently with *Figure 7*, FMNL1-interfered cells (*Figure 8*, second column) displayed small F-actin low areas at the cIS. Moreover, synapses from YFP-FMNL1βS1086A-expressing cells (fifth column) did not exhibit the F-actin depletion at the cIS, whereas synapses from the YFP-FMNL1βS1086D-expressing cells (sixth column) exhibited wide F-actin depletion at the cIS, comparable to those observed in the control YFP⁻ and in the YFP-FMNL1βWT-expressing cells (*Figure 8*). In cells expressing the YFP-FMNL1β variants, colocalization of all the YFP-FMNL1β variants with F-actin was mainly in the F-actin-low area at the cIS or at the dSMAC. When colocalization in all Z optical sections was assessed using directly the YFP fluorescence construction instead of anti-FMNL1 signal (*Figure 8B*, second row, F-actin in blue and YFP fluorescence in green), comparable results were obtained. In addition, we analyzed the subsynaptic location of FMNL1 with respect to Ser-phosphorylated PKC substrates by using an anti-FMNL1 and an anti-phospho-Ser PKC substrate. The last antibody was validated by WB, both on endogenous FMNL1 and FMNL1β variants (*Figure 3*). Moreover, the phospho-PKC does not recognize YFP-FMNL1βS1086A or S1086D variants (*Figure 3*). In addition, when FMNL1 is interfered, the phospho-PKC does not colocalize with FMNL1 and it strongly colocalizes at the synapse with expressed YFP-FMNL1βWT (*Figure 8—figure supplement 1*) in the Jurkat cell. Taken together these results indeed demonstrate the specificity of phospho-PKC antibody and that this antibody certainly recognizes phosphorylated FMNL1β in the Jurkat cell.

Both endogenous FMNL1 in control YFP⁻ cells (*Video 4* and *Figure 8—figure supplement 1A*, first column) and YFP-FMNL1βWT in YFP-FMNL1βWT⁺ cells, (*Figure 8—figure supplement 1A*, third column) strongly and specifically colocalized with anti-phospho-Ser PKC substrate at the cIS. This is evidenced by the accumulation of white, colocalization pixels at low FMNL1 density areas located at the cIS, which is also an F-actin-low area (*Figure 8*, see MFI profiles for control YFP⁻ and YFP-FMNL1βWT⁺ cells). When colocalization at the synapse interface was assessed using directly the YFP fluorescence construction instead of anti-FMNL1 signal (*Figure 8—figure supplement 1B* second row, anti-phospho-Ser PKC substrate in blue and YFP construction in green and *Video 4*) comparable results were obtained. In addition, as expected and as a negative control, no colocalization of FMNL1 and anti-phospho-Ser PKC substrate was observed in FMNL1-interfered cells (*Figure 8—figure supplement 1* and *Video 4*, second column). Moreover, FMNL1 and anti-phospho-Ser PKC also colocalized using superresolution STED microscopy, that greatly reduces xy resolution limits down to 30 nm (*Figure 8—figure supplement 2*). Although all these data do not yet allow us to infer that FMNL1β is phosphorylated at the IS due to the resolution limits of superresolution microscopy and the possibility that another PKC substrate may be associated with FMNL1 or very close to FMNL1, in a strictly S1086-dependent manner, the results are compatible with the idea that both endogenous FMNL1 and YFP-FMNL1βWT are specifically phosphorylated at the cIS.

## S1086 phosphorylation of FMNL1β regulates exosome secretion

The aforementioned results support that FMNL1β and its phosphorylation at S1086 are involved in MTOC/MVB polarized traffic to the IS, most probably via control of F-actin rearrangement at the IS. MVB transport and fusion to the plasma membrane are necessary for exosome secretion (*Théry et al., 2009*; *Calvo and Izquierdo, 2020*) and PKCδ controls F-actin depletion at the cIS (*Herranz et al., 2019*), thus, it is conceivable that FMNL1β may control subsequent exosome secretion at the IS. To directly analyze FMNL1β contribution to exosome secretion, we used an exosome secretion reporter assay. We transiently co-transfected C3 clone with the exosome reporter GFP-CD63 expression plasmid (*Alonso et al., 2011*; *Trajkovic et al., 2008*) and the different shFMNL1-YFP constructs. Subsequently, IS were formed between the co-transfected C3 clone and SEE-pulsed

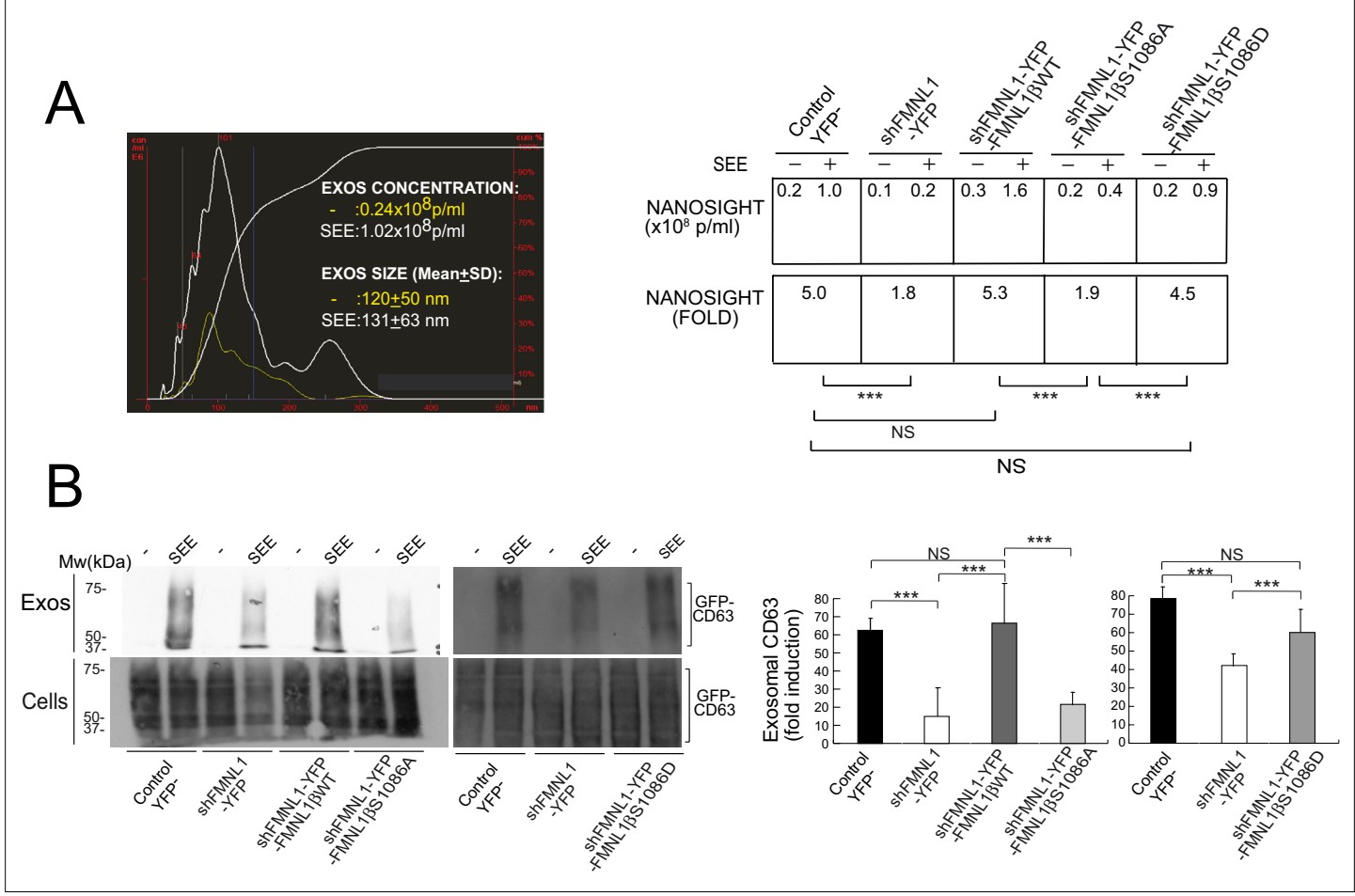

**Figure 9.** FMNL1β phosphorylation at S1086 regulates exosome secretion at the immune synapse (IS). C3 control clone was transfected with exosome reporter GFP-CD63 alone (Control YFP⁻) or co-transfected with the exosome reporter GFP-CD63 and threefold molar excess of either FMNL1-interfering (shFMNL1-HA-YFP), or FMNL1-interfering expressing interference-resistant YFP-FMNL1βWT(shFMNL1-HA-YFP-FMNL1βWT), YFP-FMNL1βS1086A, or YFP-FMNL1βS1086D constructs. Subsequently, the different cell groups were challenged with unpulsed (-) or SEE-pulsed Raji cells for 4 hr and exosomes were collected and purified from cell culture supernatants and analyzed by nanoparticle tracking analyses (NTA) and western blot (WB), as indicated in Materials and methods. (**A**) Left, NTA corresponding to nanovesicles isolated from cell culture supernatants of control YFP⁻ Jurkat cells stimulated with unpulsed Raji cells (-) or SEE-pulsed Raji cells (SEE). Right, concentration and SEE-stimulated nanoparticle concentration increase (fold) induction of secreted nanovesicles from several NTA for the indicated cell groups. (**B**) Left, WB analyses of exosomal GFP-CD63 reporter isolated from the indicated cell culture supernatants. Right, quantification of normalized, fold induction of exosomal GFP-CD63 secretion from three experiments similar to the one described in the left panel. NS, not significant; ***, p≤0.05. Results and ANOVA are representative of data from several independent experiments (n=3) with similar results.

The online version of this article includes the following video and source data for figure 9:

**Source data 1.** Original, uncropped WBs corresponding to the panel B in *Figure 9* in PDF format.

**Source data 2.** Original, uncropped WBs corresponding to the upper panel B in *Figure 9* in TIF format.

**Figure 9—video 1.** Immune synapse (IS) formation and polarized traffic of multivesicular bodies (MVB).

https://elifesciences.org/articles/96942/figures#fig9video1

GFP-CD63-expressing Jurkat C3 control clone cells were mixed with SEE-pulsed, CMAC-labeled Raji cells (blue) attached to slides to induce IS formation as indicated in Materials and methods. The video (7 fps) shows an already established synapse (center) and an emerging one (left), along with the movement of GFP-CD63⁺ MVB in C3 cells toward the synaptic contact areas. CMAC (blue) and GFP-CD63 (green) merged channels are shown. The video is a representative example out of the 21 recorded synapses. Cell surface GFP-CD63 is observed, due to fusion of GFP-CD63⁺ MVB with the plasma membrane (*Alonso et al., 2011*). This video is related to *Figure 9*.

Raji cells, exosomes were purified from cell culture supernatants (*Alonso et al., 2011*), lysed, and exosome lysates were analyzed by anti-CD63 WB. This secretion assay excludes the detection of exosomes released by Raji cells in the co-culture, that otherwise may mask the exosomes released by Jurkat cells (*Alonso et al., 2011*; *Calvo and Izquierdo, 2020*). In parallel, we determined nanovesicle concentrations in co-culture supernatants by using nanoparticle tracking analyses (NTA) (*Figure 9A*). Anti-CD63 WB analysis of exosome secretion is shown in *Figure 9B*. We normalized exosomal GFP-CD63 signal to the transfection efficiency of the exosome reporter (measured by GFP-CD63 signal from cell lysates) and the number of viable exosome-secreting cells (*Figure 9B*). As shown in *Figure 9*, interference with all FMNL1 isoforms not only reduced exosomal GFP-CD63 levels, but also reduced SEE-stimulated nanovesicle concentration increase. These results mirrored those obtained in cells in PKCδ-interfered cells (*Herranz et al., 2019*). Moreover, interference with all FMNL1 isoforms and YFP-FMNL1βWT expression recovered exosome secretion to the levels produced by control YFP⁻ cells. However, YFP-FMNL1βS1086A expression did not recovered exosome secretion, whereas YFP-FMNL1βS1086D expression rescued exosome secretion to the values observed in control YFP⁻ or YFP-FMNL1βWT-expressing cells (*Figure 9B*). These results demonstrate IS-induced exosome secretion, but do not reveal the presence of exosomes at the synaptic cleft. Therefore, we have performed STED superresolution imaging of the IS made by control and FMNL1-interfered cells. Nanosized (100–150 nm) CD63⁺ vesicles can be found in the synaptic cleft between the APC and control YFP⁻ cells with polarized MVB, whereas we could no detect these vesicles in the cleft from FMNL1-interfered cells that maintain unpolarized MVB (*Figure 10*). Taken together, these results on exosome secretion endorse the outcomes regarding FMNL1 interference and FMNL1β-mediated control of cortical F-actin reorganization at the IS and their role on MVB polarized secretory traffic.

## Discussion

Our previously published results suggest that PKCδ-dependent phosphorylation of FMNL1β may regulate its function in F-actin reorganization at the IS and hence affect MTOC polarization (*Herranz et al., 2019*; *Bello-Gamboa et al., 2020*). We have previously shown that phosphorylation occurs in FMNL1 but not in a homologous formin, Dia1 (*Bello-Gamboa et al., 2020*), that also regulate MTOC polarization and it is a major actin regulator in T lymphocytes (*Gomez et al., 2007*), supporting an specific role of FMNL1 phosphorylation, but not Dia1 phosphorylation on MTOC polarization. Here, we have identified for the first time a positive regulatory role of FMNL1β phosphorylation at S1086 in cortical F-actin regulation that governs MTOC/MVB polarization and exosome secretion at the IS in Th lymphocytes. Moreover, the fact that the phosphomimetic YFP-FMNL1βS1086D expression in PKCδ and FMNL1-interfered cells did not recover the MTOC PI observed in C3 control cells (*Figure 5*) supports the idea that FMNL1β phosphorylation at S1086 is necessary but not sufficient for MTOC polarization, at least in cells lacking PKCδ. Thus, these results support that another signal, apart from PKCδ activation, or another PKCδ-controlled pathway not involving FMNL1β phosphorylation, evoked upon TCR triggering at the IS, is also implicated in MTOC/MVB polarization to the IS.

Our results on FMNL1β phosphorylation are certainly related to what was found in formin homology domain protein 1 (FHOD1) regulation. FHOD1 is grouped into Dia-related formins (*Schönichen and Geyer, 2010*). FHOD1 requires active Rac to be recruited to the plasma membrane, but its recruitment appears insufficient for FHOD1 activation, which is achieved only after being phosphorylated by Rho-dependent protein kinase (ROCK) at three serine and threonine sites included in a polybasic arginine-rich region inside the C-terminal DAD (*Takeya et al., 2008*), leading to disruption of FHOD1 autoinhibitory state and F-actin stress fiber formation (*Takeya et al., 2008*). This activation process is similar to what we have described in this article for FMNL1β. Moreover, the PKC-dependent mode of FMNL2 activation during integrin activation (*Wang et al., 2015*) and filopodia formation (*Lorenzen et al., 2023*) is comparable to what we have found in FMNL1β during polarized traffic to the IS. Furthermore, FHOD3 formin is also autoinhibited by an intramolecular interaction between its C- and N-terminal domains. Phosphorylation of the three highly conserved residues (S1406, S1412, and T1416) within the polybasic, C-terminus of FHOD3 by ROCK1/2 is sufficient for its activation and is crucial in regulating myofibrillogenesis in cardiomyocytes (*Zhou et al., 2017*). Thus, with the results provided here for FMNL1β, it has been reported that at least four different formins are activated through phosphorylation by three different kinases at the C-terminal polybasic regions, which are included in a common DAD. This underscores the evolutionary importance of this regulatory mechanism in

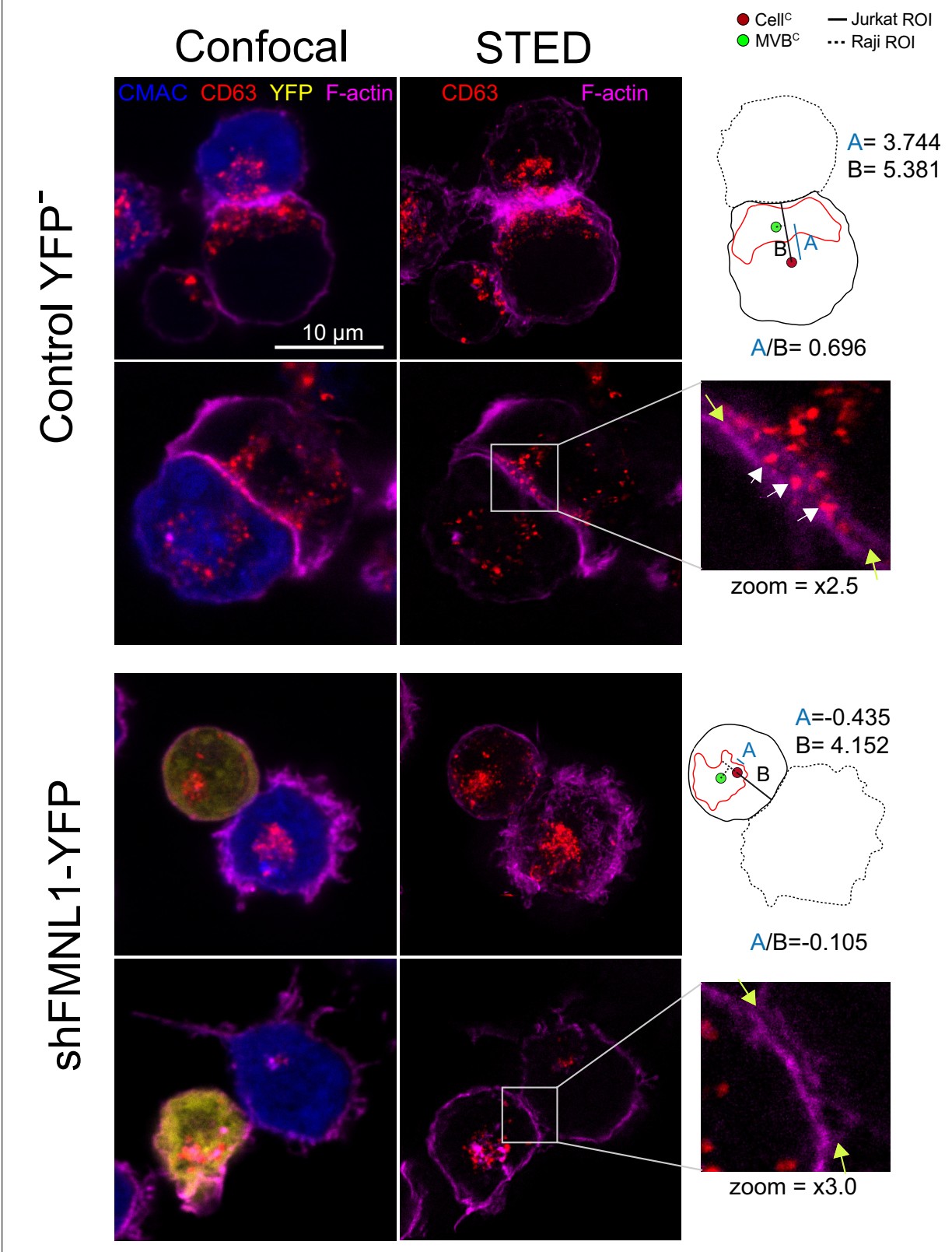

**Figure 10.** STED image of CD63⁺ nanovesicles at the synaptic cleft. C3 control clone cells untrasfected or expressing shFMNL1-HA-YFP were challenged with CMAC-labeled SEE-pulsed Raji cells (blue) for 1 hr, fixed, stained with phalloidin AF647 (magenta) and anti-CD63 (red) and imaged simultaneously by confocal and STED microscopy. Two representative control YFP⁻ and FMNL1-interfering (shFMNL1-HA-YFP) Jurkat cells forming immune synapse (IS) with Raji cells are shown. Confocal (left) and STED (right) optical sections are shown, and enlarged (2.5× and 3× zoom) views of

*Figure 10 continued on next page*

*Figure 10 continued*

the IS areas are shown in the right side. The yellow arrows on the enlarged IS images label the edges of the synaptic cleft, which is the narrow, lane-shaped space between the two cells enclosed by the two F-actin-rich (magenta) plasma membrane leaflets. CD63+ multivesicular bodies (MVB) from the Jurkat cells are located nearby to the IS and some CD63+ nanovesicles (white arrows) are located at the synaptic cleft in the control YFP- example. The diagrams used to calculate the MVB polarization index (PI) data in both cell groups are represented in the right side of the images. The zoom of the indicated regions of interest (ROIs) (white square) is included below the diagrams. The percentage of synaptic conjugates on which was evident the presence of CD63+ nanovesicles at the synaptic cleft was 17% for control YFP-, and 0% for FMNL1-interfering (shFMNL1-HA-YFP), respectively, with at least 35 synapses analyzed per condition. Images are representative of the data from several independent experiments (n=3) with similar results.

formin activation, ultimately leading to very diverse cellular responses. However, although we do not have conclusive evidence supporting that PKC directly phosphorylates FMNL1β, the presence of a conserved S1086 within an RRSV/AR motif that contains a potential target for classical PKCs (*Wang et al., 2015*; *Lorenzen et al., 2023*; *Nishikawa et al., 1997*), along with the recognition of the phosphorylated residue by a validated anti-Phospho-Ser PKC substrate antibody (*Wang et al., 2015*), certainly support this possibility.

With regard to potential regulatory pathways involved in formin activation and subcellular localization, it is remarkable that the most abundant formins in T lymphocytes, Dia1 and FMNL1 (*Gomez et al., 2007*), are constitutively inactive in the cytoplasm due to intramolecular DAD-DID binding, which blocks formin ability to nucleate and elongate actin filaments (*Hammer et al., 2019*). Regarding a possible control of formin subcellular location by PKC, it has been shown that both relocation of FMNL2 from the plasma membrane to the endosomal compartment and activation are dependent on C-terminal S1072 phosphorylation by PKCα (*Wang et al., 2015*). However, FMNL2 location in filopodia seems to be independent of the phosphorylation state (*Lorenzen et al., 2023*). We have previously described PKCδ-dependent phosphorylation of FMNL1β (*Bello-Gamboa et al., 2020*), and here we have demonstrated that this phosphorylation occurs at S1086, leading to F-actin reorganization at the IS, and resulting in MTOC polarization. However, in contrast to FMNL2 relocation, which is dependent on PKCα, we have shown that transient YFP-FMNL1βWT relocation from cytosol to the IS occurred in the absence of PKCδ (*Figure 6*), in conditions where neither F-actin reorganization (*Herranz et al., 2019*) nor MTOC polarization (*Figure 5*) occurred. Moreover, both YFP-FMNL1βS1086A and YFP-FMNL1βS1086D translocated to the IS (*Video 3* and *Figure 6—figure supplement 2*), demonstrating that S1086 phosphorylation is not required for IS location. However, as it happens during S1072 phosphorylation in FMNL2 (*Wang et al., 2015*), it is possible that blocking phosphorylation at S1086 through the introduction of single point mutations (YFP-FMNL1βS1086A or YFP-FMNL1βS1086D) may not block FMNL1β translocation to the IS per se, but may alter the kinetics of the process. For this reason, in future approaches, a thorough study on the kinetic considerations on S1086 PKCδ-mediated phosphorylation contribution to FMNL1β translocation to the IS will need to be addressed.

Jurkat T cells may not fully recapitulate the regulation of polarization in primary T cells, and the evidences from activating surfaces acting as artificial synapses show distinct actin cytoskeleton behavior (presence or not of F-actin foci) in primary versus immortalized T cells, at least in CD4+ cells, that may reflect mechanistic distinctions in the T cell models systems utilized to study cytoskeletal actin dynamics (*Kumari et al., 2019*; *Colin-York et al., 2019*). However, to date, no clear evidence has been obtained regarding the existence of such an F-actin foci network in cell-cell IS models based on primary T lymphocytes or Jurkat cells. F-actin foci have been exclusively observed in primary T cells engaged with artificial activating surfaces (planar lipid bilayers or coated glass), which does not apply to our data obtained from cell-cell synapses. However, to solve any potential differences in FMNL1 localization between primary versus immortalized T cells, we have performed experiments analyzing the subcellular localization of FMNL1 in CD19/CD22 dual-targeted CAR T cells forming synapses with CD19+/CD22+ Raji cells used as CTL targets and T lymphoblasts forming synapses with SEE and SEB-pulsed Raji cells. We observed, in fixed-cell images, the presence of endogenous FMNL1 colocalizing with F-actin-enriched areas in the IS region of both CAR T cells and primary T lymphoblasts, which also exhibited MTOC polarization toward the IS (*Figure 6—figure supplement 1*). The presence of FMNL1 at the IS developed by primary T lymphoblasts was transient, as occurred in Jurkat cells (not shown). These observations extend our results on FMNL1 localization at the IS in the Jurkat-Raji plus SEE synapse model to the IS developed by primary human T lymphocytes. Regarding exosome secretion in primary human T lymphocytes, additional experiments involving the expression of the

diverse FMNL1β variants will be necessary to assess whether FMNL1β and its phosphorylation at S1086 controls exosome secretion in primary T lymphocytes, as occurred in Jurkat cells. These experiments will require the development of new transduction protocols to improve the otherwise inefficient expression in primary T lymphocytes, that it is even worse using the bi-cistronic large plasmids (>15 kb) that we have employed in our studies. In addition, it is well known that the mere expression of large proteins such as the 180 kDa (30+150 kDa) YFP-FMNL1 chimeric variants constitutes a significant challenge. Thus, we have been unable to achieve enough transfection efficiency to perform these experiments, which would have allowed us to fully extend the effect of FMNL1β and its phosphorylation at S1086 on exosome secretion to primary T cells. However, FMNL1 is necessary for MTOC/MVB polarization in Jurkat cells (*Gomez et al., 2007*; *Bello-Gamboa et al., 2020*), as confirmed in this manuscript, and this finding has already been extended to primary CD8[+] T cell clones (*Gomez et al., 2007*). All of this, together with the fact that exosome secretion requires MTOC/MVB polarization both in Jurkat cells and primary T lymphoblasts (*Alonso et al., 2011*; *Bello-Gamboa et al., 2020*), suggests FMNL1 may also control exosome secretion in primary T cells, although the formal demonstration will require further research. Regarding the effect of the diverse FMNL1β variants expression on MTOC/MVB polarization and exosome secretion in primary T lymphocytes, the data from activating surfaces clearly shows that the synaptic actin architecture of the IS from primary CD8[+] T cells is essentially indistinguishable and thus unbiased from that of Jurkat T cells, but different to that of primary CD4[+] cells (*Murugesan et al., 2016*). Thus, it is expected that all our data in Jurkat T cells could probably be extrapolated to the synaptic architecture of primary CD8[+] cells, although more experiments involving primary CD4[+] are necessary to extend all our results to the last cell type.

Although the mechanism(s) involved in FMNL1β recruitment to the IS remains unclear, it is probably that it involves Rac1 and Cdc42 small G proteins (*Kühn and Geyer, 2014*; *Gomez et al., 2007*; *Favaro et al., 2013*), which are activated at the IS upon TCR triggering (*Deckert et al., 2005*). However, we cannot exclude the participation of another PKC isoform or a different protein kinase that may induce FMNL1β phosphorylation in different residues upon previous FMNL1β relocation to the IS primed by small G protein, as was shown for FMNL2 relocation (*Wang et al., 2015*). Further experiments are needed to establish this important point.

Taken together, our results support that IS-induced, PKCδ-dependent phosphorylation of S1086 in FMNL1β DAD, most probably executed by PKCδ itself, activates FMNL1β, which in turn regulates cortical actin at the IS and hence controls MTOC/MVB polarization and exosome secretion. The behavior of non-phosphorylatable YFP-FMNL1βS1086A (equivalent to FMNL2-S1072A described in *Wang et al., 2015*) confirms this hypothesis, since its expression inhibits F-actin reorganization at the IS, MTOC polarization, and exosome secretion. In this respect, it has been reported that the phosphorylation of FMNL2 by PKCα in S1072 is necessary for the secretion of the pro-metastatic factor Angiopoietin-like 4 (ANGPTL4) in breast cancer cells (*Frank et al., 2023*), which constitutes the first description for an activated formin involvement in secretion during cell invasion. In addition, ANGPTL4 can be secreted into exosomes by lung cancer cells (*Mo et al., 2020*; *Zhang et al., 2022*) and FMNL2 controls exosome secretion (*He et al., 2023*). Thus, although it is not clear whether ANGPTL4 is released in exosomes or in soluble form (*Frank et al., 2023*), both FMNL2 and FMNL1β appear to control PKC-regulated exosome secretion. However, the extracellular signals evoking exosome secretion are strikingly different in the various cell types, since in tumor cells hypoxia and radiation can amplify the otherwise constitutive and multidirectional exosome secretion, whereas T lymphocytes constitutively secrete very low levels of exosomes and none in a polarized manner unless activated through TCR or synaptic contact (*Calvo and Izquierdo, 2020*). Taken together, these considerations support a more general role of the formin-actin cytoskeleton axis in directional exosome secretion. Given all these considerations, it will be interesting to analyze the role of FMNL2 in inducible exosome secretion at the IS in T lymphocytes, but also the role of FMNL1β in exosome secretion by cancer cells during cell invasion. This is particularly relevant as exosome secretion is required for directionally persistent and efficient in vivo movement of cancer cells (*Sung et al., 2015*), and it is known that FMNL1 stimulates both leukemia cell proliferation and migration (*Favaro et al., 2013*; *Thompson et al., 2020*).

The fact that FMNL1- and PKCδ-interfered cells have a similar phenotype to that of FMNL1- or PKCδ-interfered cells (*Bello-Gamboa et al., 2020*; *Herranz et al., 2019*) (this paper) supports the notion that PKCδ and FMNL1β participate in the same F-actin regulatory pathway controlling MTOC/

MVB polarization. Additional experiments involving in vitro measurements of F-actin reorganizing/severing activities of the diverse FMNL1β variants, phosphorylated or not in vitro by PKCδ, will be necessary to address whether PKCδ directly phosphorylates FMNL1β at S1086.

Our findings show that FMNL1 colocalizes with F-actin at the IS edges which are part of the dSMAC (*Figures 6 and 8*). These data are aligned with the previous observation that Dia1 and FMNL1 are intrinsically inactive because they undergo intramolecular folding and remain in the cytoplasm (*Hammer et al., 2019*) and that after IS formation, they are enriched at the tips of F-actomyosin spikes at the outer edge of the dSMAC, giving rise to the actomyosin arcs at the pSMAC in the IS formed by Jurkat cells (*Murugesan et al., 2016*; *Hammer et al., 2019*). Dia1 is not a target of PKC, whereas FMNL1 is phosphorylated by PKC activation (*Bello-Gamboa et al., 2020*). Thus, in this paper we have focused on FMNL1. High-resolution, spatiotemporal analysis of FMNL1β localization at the different subsynapic locations, together with superresolution techniques (*Murugesan et al., 2016*; *Calvo and Izquierdo, 2018*), may provide new clues regarding the subsynaptic F-actin network targeted by phosphorylated FMNL1β. Even though the nature of the downstream effectors and consequences of FMNL1β phosphorylation on synaptic F-actin architecture can only be speculated at this point, concentric actomyosin arcs generated by FMNL1 propel TCR microclusters toward the IS and reinforce cytotoxicity (*Murugesan et al., 2016*; *Hammer et al., 2019*). Thus, high-resolution spatiotemporal analysis of actomyosin arcs structure and function during TCR microcluster movement in FMNL1-interfered cells, and/or cells expressing the FMNL1β mutants described here, will provide important clues on this important matter. It is noteworthy that a majority of the image analysis of IS organization and F-actin dynamics is based on planar synapses obtained using artificial APC substitutes, by either coating stimulatory and co-stimulatory molecules on glass or plastic surfaces or by embedding these molecules into lipid bilayers (*Dustin, 2009*; *Hammer et al., 2019*; *Dupré et al., 2021*). These artificial synapses can indeed be imaged and analyzed with the highest possible resolution (*Carisey et al., 2018*; *Murugesan et al., 2016*; *Calvo and Izquierdo, 2018*). However, the cell-cell synapse model analyzed here mimics the complex interactions and irregular, 3D stimulatory surface of a physiological synapse better than artificial synapses do (*Dustin, 2009*; *Hammer et al., 2019*). In this important sense, the data presented here are indeed closer to a physiological situation. In addition, since acto-myosin arcs arise from active, formin-dependent nucleation sites at the outer edge of the dSMAC (*Murugesan et al., 2016*), it will be interesting to analyze, using superresolution techniques, whether phosphorylated FMNL1β is located at these sites. Indeed, actomyosin arc network spanning from dSMAC along the pSMAC is the most likely source of T cell-based force that augments cytotoxicity by straining the target cell plasma membrane (*Hammer et al., 2019*). Thus, it will be interesting to study the effect of FMNL1β mutants on actomyosin arcs formation and analyze functional deficiencies (i.e. decreased T cell-APC adhesion frequency, IS size, and IS formation efficiency) in IS formed by cells interfered in FMNL1β and/or expressing the FMNL1β variants described here. Visualization of actin and actomyosin networks, TCR microclusters, and secretion granules dynamics in 'real' live T cell-APC conjugates at the superresolution level (*Hammer et al., 2019*) is clearly needed to advance in this area.

Remarkably, although it has already been shown that FMNL1 is important for MTOC polarization in Jurkat cells and for target cell lysis in CTL (*Gomez et al., 2007*), our data provide the first molecular basis underpinning this essential FMNL1 function. Here, we demonstrate that FMNL1β and its phosphorylation at S1086 participate in F-actin-controlled, polarized MVB traffic secreting effector exosomes in Jurkat T lymphocytes developing IS (*Alonso et al., 2011*; *Alonso et al., 2005*), which deliver pro-apoptotic signals to target cells (*Bálint et al., 2020*; *Chang et al., 2022*; *Cassioli and Baldari, 2022*). In addition, FMNL1 promotes proliferation and migration of leukemia cells (*Favaro et al., 2013*) and mediates posterior perinuclear actin polymerization to promote T lympho-cyte effector cell migration to inflammatory sites to enable T cell-mediated autoimmunity (*Thompson et al., 2020*). PKCδ is crucial for directional T cell migration (*Fanning et al., 2005*; *Volkov et al., 1998*) and this process requires MVB polarized traffic and exosome secretion in the direction of migration (*Sung et al., 2015*). For that reason, it will be interesting to analyze the effects of the FMNL1β mutants we have described here during T lymphocyte migration and tumorigenesis, as well as to study the role of FMNL1β phosphorylation in several autoimmune disorders. In addition, improving our understanding of the molecular bases underlying the traffic events involved in polarized secretion of pro-apoptotic exosomes as we have performed here, provided that the FMNL1 effect on exosome

secretion in Jurkat cells can be extended to primary T lymphocytes, will deliver clues to modify crucial immune functions involving apoptosis, such as cytotoxicity by CTLs and immunoregulatory AICD and their associated pathologies. This may allow to design therapeutic strategies to modify CAR T lifespan and/or polarized secretory function, in order to improve their efficiency for the treatment of certain cancers (*Alonso et al., 2005*; *Alonso et al., 2007*; *Alonso et al., 2011*; *Mazzeo et al., 2016*; *Herranz et al., 2019*; *Stinchcombe et al., 2004*; *de Saint Basile et al., 2010*; *Davenport et al., 2018*).

## Materials and methods

### Cells

Raji B cell line was obtained from the ATCC. Cell lines were cultured in RPMI 1640 medium containing L-glutamine (Invitrogen) with 10% heat-inactivated FCS (Gibco) and penicillin/streptomycin (Gibco). Jurkat control (C3 and C9) and PKCδ-interfered (P5 and P6) stable clones derived from JE6.1 clone have already been described and were cultured with puromycin (*Herranz et al., 2019*). C3 and C9 control clones are used, instead of canonical JE6.1 cells, since these puromycin-resistant control clones (containing a scramble shRNA) were isolated by limiting dilution together with the PKCδ-interfered clones (*Herranz et al., 2019*), thus are the best possible controls for P5 and P6 clones. The P5 and P6 clones exhibited a comparable defect in MVB/MTOC polarization when compared with the control clones (*Herranz et al., 2019*; *Bello-Gamboa et al., 2020*). In addition, when interference-resistant GFP-PKCδ was transiently expressed in all the PKCδ-interfered clones, MTOC/MVB polarization was recovered to control levels (*Herranz et al., 2019*). Therefore, the deficient MTOC/MVB polarization in these clones is exclusively due to the reduction in PKCδ expression (*Herranz et al., 2019*), and thus clonal variation cannot be responsible for the results in stable clones. Since the FMNL1 interference and YFP-FMNL1 variants re-expression experiments are performed in transient assays (2–4 days after transfection), there is no chance for any clonal variation in these short-time experiments. Peripheral mononuclear cells from healthy donors (La Paz University Hospital, Madrid, Spain) were stimulated with phytohemagglutinin (PHA-M, 1 µg/ml) for 3 days. Dual CAR T cells recognizing CD19/CD22 were produced as previously shown (*Ibáñez-Navarro et al., 2023*) and were provided by Department of Pediatric Hemato-Oncology, La Paz University Hospital (Madrid, Spain). Subsequently, both T lympho-blasts and CAR T cells were expanded in the presence of recombinant human IL-2 (20 ng/ml, Roche).

### Plasmids and transient transfection

pEGFP-C1CD63 (expressing GFP-CD63) was provided by G Griffiths. CD63 is an enriched marker in intraluminal vesicles (ILVs) contained into MVB and exosomes and has been widely used to study MVB secretory traffic in both living and fixed cells (*Alonso et al., 2011*; *Alonso et al., 2005*). The FMNL1-interfering vector (shFMNL1-HA-YFP) and FMNL1-interfering, YFP-FMNL1βWT expressing vector (shFMNL1-HA-YFP-FMNL1βWT) were previously described (*Colón-Franco et al., 2011*) and generously provided by Dr. Billadeau. shFMNL1-HA-YFP-FMNL1βS1086A and shFMNL1-HA-YFP-FMNL1βS1086D mutants were generated by site-directed mutagenesis, as previously reported (*Barbeito et al., 2021*). Briefly, overlap extension PCR was used to introduce the desired muta-tions into a 900 bp SalI-NotI fragment, which was then used to replace the corresponding wild-type sequence in shFMNL1-HA-YFP-FMNL1β-WT. All amplifications were carried out with Platinum SuperFi DNA polymerase (Thermo Fisher). Primers used were: SalI-FMNL1β_F (CCAGAGCCTGGATGCG CTGTTGG), FMNL1β-S1086A_F (CAGACACAGGCCGCCGCGCTGCCCGTCGGCGTCCC), FMNL1β-S1086A_R (GGGACGCCGACGGGCAGCGCGGCGGCCTGTGTCTGC), FMNL1β-S1086D_F (CAGA CACAGGCCGCCGCGATGCCCGTCGGCGTCCC), FMNL1β-S1086D_R (GGGACGCCGACGGGCA TCGCGGCGGCCTGTGTCTGC), and NotI-FMNL1β_R (GCGAGCTCTAGGGCCGCTTGCG). Presence of the desired mutations, and absence of undesired ones, was confirmed by Sanger DNA sequencing (Eurofins Genomics, LightRun). Both interference and expression of the different chimeric molecules were assessed in single cell level by immunofluorescence and/or bulk cell populations by WB with an anti-FMNL1 antibody, which recognizes all FMNL1 isoforms (see below). Jurkat clones were tran-siently transfected with 20–30 µg of the plasmids as described (*Alonso et al., 2005*). To analyze the effects of shFMNL1 constructs on exosome secretion by Jurkat cells expressing exosome reporter GFP-CD63, Jurkat cells were transiently co-transfected with a threefold molar excess of the different shFMNL1-YFP plasmids with respect to the GFP-CD63 exosome reporter plasmid, to favor that the

exosomes analyzed were mainly those secreted by Jurkat cells containing both the GFP-CD63 plasmid and the shFMNL1 construct (*Alonso et al., 2011*; *Mazzeo et al., 2016*).

## Antibodies and reagents

Rabbit monoclonal anti-human PKCδ EP1486Y for WB that does not recognize mouse PKCδ was from Abcam. Rabbit monoclonal anti-PKCδ EPR17075 for WB that recognizes both human and mouse PKCδ was from Abcam. Mouse monoclonal anti-human CD3 (clone UCHT1) for cell stimulation and immunofluorescence was from BD Biosciences and Santa Cruz Biotechnology. Mouse monoclonal anti-FMNL1 clone C-5 for WB and immunofluorescence and mouse monoclonal anti-FMNL1 clone A-4 for immunoprecipitation were from Santa Cruz Biotechnology, recognizing all FMNL1 isoforms. Mouse monoclonal anti-γ-tubulin for immunofluorescence was from SIGMA, (clone GTU-88). Rabbit polyclonal anti-pericentrin ab4448 for immunofluorescence was from Abcam. Mouse monoclonal anti-human CD63 (clone TEA3/18) for immunofluorescence was from Immunostep. Mouse monoclonal anti-CD63 clone NKI-C-3 for WB was from Oncogene. Rabbit polyclonal Phospho-Ser PKC substrate antibody for WB and immunofluorescence was from Cell Signaling Technology. Fluorochrome-coupled secondary antibodies (goat anti-mouse IgG AF488 A-11029, goat anti-rabbit IgG AF488 A-11034, goat anti-mouse IgG AF546 A-11030, goat anti-mouse IgG AF647 A-21236) for immunofluorescence were from Thermo Fisher. Horseradish peroxidase (HRP)-coupled secondary antibodies (goat anti-mouse IgG-HRP, sc-2005 and goat anti-rabbit IgG-HRP, sc-2004) for WB were from Santa Cruz Biotechnology. CellTracker Blue (CMAC) and phalloidin were from Thermo Fisher. Staphylococcal enterotoxin E (SEE) was from Toxin Technology, Inc Phytohemagglutinin (PHA-M) was from SIGMA. Recombinant human IL-2 was from Roche.

## Immunoprecipitation

Immunoprecipitation from cell lysates was performed by using Protein A/G Magnetic Beads (Pierce, Thermo Scientific) following the instructions provided by the company. Briefly, 0.5 ml lysates corresponding to $2–4×10^6$ transfected Jurkat cells, stimulated or not with PMA (100 ng/ml, 30 min at 37°C), were incubated with anti-FMNL1 (clone A4, 5 µg) for 2 hr at 4°C. Subsequently, 15 µl of magnetic beads suspension was added and incubated for 3 hr at 4°C. Beads were washed 5× with lysis buffer and the antigens were eluted with 2 M glycine pH = 2 and then neutralized. Eluates were run on 6.5% SDS-PAGE gels and proteins transferred to PVDF membranes.

## Isolation and quantification of exosomes

To analyze the exosomes produced by cells transfected with the exosome reporter GFP-CD63, a similar protocol to the originally described for exosome isolation (*Théry et al., 2006*) was performed. We have included a 48 hr post-transfection Ficoll gradient purification step to remove dead cells and cell debris, which otherwise could contaminate exosome preparations with micro- and nanoparticles (*Alonso et al., 2011*; *Mazzeo et al., 2016*). Using these standard protocols, culture supernatants from Jurkat cells transfected with GFP-CD63 or co-transfected with GFP-CD63 and with a threefold molar excess of the different shFMNL1-YFP plasmids, co-cultured with SEE-pulsed Raji cells, were centrifuged in sequential steps to eliminate cells and cell debris/apoptotic bodies (*Théry et al., 2006*). Subsequently, the exosomes were recovered by ultracentrifugation (100,000×$g$ for 12 hr at 4°C) as described (*Martínez-Lorenzo et al., 1999*), lysed and then analyzed by WB. CD63 is characteristically present in MVB, ILVs, and hence in exosomes, but also in secretory lysosomes and the plasma membrane (*Figure 9—video 1*). Plasma membrane CD63 localization is produced by degranulation of MVB and diffusion of CD63 from the limiting membrane of MVB to the plasma membrane upon MVB fusion (*Alonso et al., 2011*; *Mazzeo et al., 2016*; *Figure 9—video 1*). Approximately $1–2×10^6$ transfected Jurkat cells were challenged with SEE-pulsed Raji cells and WB signals in exosome lysates were normalized by the cell expression levels of GFP-CD63 among different transfections, cell groups, and stimuli, in the WB corresponding to the cell lysates (*Alonso et al., 2011*; *Trajkovic et al., 2008*; *Mazzeo et al., 2016*). It has been established that the exosomal CD63 WB signal in this synapse model correlates with exosome number obtained by flow cytometry (*Ostrowski et al., 2010*), by electron microscopy (*Ventimiglia et al., 2015*), and by nanoparticle concentration analysis (nanoparticles/ml), using NTA (*Mazzeo et al., 2016*). WB analysis of CD63 protein and GFP-tagged CD63 in isolated exosomes has been used as a bona fide method to specifically determine changes in

exosome production (*Alonso et al., 2005*; *Trajkovic et al., 2008*; *Raiborg et al., 2003*) in transiently transfected Jurkat cells forming synapses with SEE-pulsed Raji cells without the detection of contaminating exosomes produced by MHC-II-stimulated Raji cells or constitutively secreted by Raji cells (*Muntasell et al., 2007*; *Alonso et al., 2011*; *Alonso et al., 2005*; *Théry et al., 2006*; *Mazzeo et al., 2016*; *Herranz et al., 2019*; *Calvo and Izquierdo, 2020*). No significant differences in the GFP-CD63 levels (i.e. *Figure 9B*) were observed in the lysates of transfected C3 Jurkat cells, stimulated or not with SEE-pulsed Raji cells, at the end of the cell co-culture period for exosome secretion, showing that the GFP-CD63$^+$ exosomes were produced by an equal number of viable, GFP-CD63-expressing cells. Moreover, to quantify exosome concentration and to analyze their size distribution, the cell culture supernatant collected just before the ultracentrifugation step was diluted (1/5) in Hank's balanced salt solution and analyzed by NTA using a NANOSIGHT equipment (LM10, Malvern) that was calibrated with 50 nm, 100 nm, and 400 nm fluorescent calibration beads (Malvern). The hydrodynamic diameter measured by NTA, although apparently higher to that originally described for exosomes using electron microscopy (50–100 nm) (*Figure 9A*), certainly corresponds to the real size of canonical, unfixed exosomes in solution, as described (*Skliar et al., 2018*). The NTA measurements of exosome concentration (particles/ml) were normalized by the transfection efficiency and exosome-producing Jurkat cell number, by referring exosome concentration to GFP-CD63 signals in the WB of the cell lysates (*Figure 9B*).

## WB analysis

Cells were lysed in Triton X-100-containing lysis buffer supplemented with both protease and phosphatase inhibitors. Approximately 50 µg of cellular proteins were recovered in the 10,000×*g* pellet from 10$^6$ cells. Cell lysates and neutralized, acid-eluted IPs were separated by SDS-PAGE under reducing conditions and transferred to Hybond ECL membranes (GE Healthcare). Membranes were incubated sequentially with the different primary antibodies and developed with the appropriate HRP-conjugated secondary antibody using enhanced chemiluminescence (ECL). When required, blots were stripped following standard protocols prior to reprobing them with primary and HRP-conjugated secondary antibodies. For exosome secretion studies, cells and isolated exosomes were lysed in RIPA lysis buffer containing protease inhibitors. Approximately 5 µg of exosomal proteins were recovered in the 100,000×*g* pellet from 1 to 2×10$^6$ cells. Exosomes were resuspended in 60 µl of RIPA lysis buffer and 20 µl of exosomal or cell lysate proteins were separated by SDS-PAGE and transferred to Hybond ECL membranes (GE Healthcare). For CD63 detection, proteins were separated under non-reducing conditions as described (*Alonso et al., 2005*). For WB analysis of exosomes, each lane contained the total exosomal protein that was recovered from the culture medium coming from the same number of cells, untreated or treated with stimuli. Blots were incubated with mouse anti-CD63 (clone NKI-C-3, Oncogene) and developed with the appropriate HRP-conjugated secondary antibody using ECL. Autoradiography films were scanned and the bands were quantified using Quantity One 4.4.0 (Bio-Rad) and ImageJ (Rasband, W.S., ImageJ, National Institutes of Health, Bethesda, MD, USA, http://rsb.info.nih.gov/ij/, 1997–2004) software.

## Time-lapse microscopy, immunofluorescence, and image analysis

In the experiments requiring IS formation, we challenged the transfected Jurkat clones with SEE-pulsed Raji cells, a well-established synapse model (*Montoya et al., 2002*). Raji cells were attached either to ibiTreat microwell culture dishes (ibidi) pretreated with fibronectin (0.1 mg/ml, for paraformaldehyde [PFA] fixation) or glass bottom microwell culture dishes (ibidi) pretreated with poly-L-lysine (0.02 mg/ml, for PFA and acetone fixation). Next, they were labeled with CMAC (10 µM) and pulsed with 1 µg/ml SEE and mixed with Jurkat clones transfected with the different expression plasmids, at 24–48 hr post-transfection. The resulting IS were analyzed as described (*Montoya et al., 2002*; *Alonso et al., 2011*; *Bello-Gamboa et al., 2019*; *Bello-Gamboa et al., 2020*). PFA followed by acetone fixation was required for a clean FMNL1 staining and was compatible with phalloidin labeling (*Abrahamsen et al., 2018*). Immunofluorescence of fixed synapses was performed as previously described (*Jambrina et al., 2003*), and additional blocking and fixations steps were performed between each primary antibody and subsequent fluorochrome-coupled secondary antibody or phalloidin staining, to exclude any potential cross-reaction of secondary antibodies (i.e. *Figure 5*).

For FMNL1β relocalization experiments in living cells, control (C3), and PKCδ-interfered (P5), YFP-FMNL1β-expressing Jurkat clones were challenged with CMAC-labeled, SEE-pulsed Raji cells as described above. Widefield, time-lapse microscopy was performed using an OKO-lab stage incubator (OKO) for sample ambient on a Nikon Eclipse TiE microscope equipped with a Prime BSI (Photometrics) digital camera, a PlanApo VC 60×/1.4NA OIL objective (Nikon) and with Perfect Focus System, allowing distinct, appropriate offsets per each XY field. Widefield fluorescence of fixed synapses was performed by capturing 30–40 Z sections (0.3–0.4 μm thickness) using the same microscope and NIS-AR software (Nikon), and maximum intensity projection (MIP) images of all channels were generated using NIS-AR software.

Time-lapse acquisition during the indicated times and analysis were performed using NIS-AR software (Nikon). Subsequently, in some experiments epifluorescence images were improved by Huygens Deconvolution Software from Scientific Volume Image, using the 'widefield' optical option as previously described (*Calvo and Izquierdo, 2018*; *Bello-Gamboa et al., 2019*). For quantification, digital images were analyzed using NIS-AR (Nikon) or ImageJ software. The quantification and analysis of YFP MFI at the synapse area (YFP-FMNL1β MFI IS) and at the whole cell (YFP-FMNL1β MFI Cell) in time-lapse experiments were performed within floating regions of interest (ROI) (i.e. ROI changing XY position over time) by using NIS-AR software (*Figure 6A* and *Figure 6—figure supplement 2*).

Confocal microscopy imaging in fixed synapses was performed by using an SP8 Leica confocal microscope (63×/1.40 Oil DIC objective), with sequential acquisition, bidirectional scanning, and the following laser lines: UV (405 nm, intensity: 33.4%), supercontinuum visible (633 nm, intensity: 15.2%), supercontinuum visible (550 nm, intensity: 20.8%), supercontinuum visible (488 nm, intensity: 31.2%). STED fluorescence microscopy was performed by using a Stellaris 8 Tau STED (Leica) equipped with a 775 nm depletion laser and a 100×/1.4 Oil STED White Objective. Deconvolution of confocal images and widefield images was performed by using Huygens Deconvolution Software from Scientific Volume Image with the 'confocal' optical option or 'widefield' option, respectively. 2D colocalization analyses were accomplished by using JACoP plugin from ImageJ, whereas interface colocalization and the corresponding videos to generate the IS interface were performed using the 'colocalization' tool, followed by the 'volume view' and then 'movie maker' tools from NIS-AR. For interface colocalization analyses, the original emission colors of the different fluorochromes were changed to red, green, or blue, since NIS-AR requires an RGB color code for this analyses (*Ruiz-Navarro et al., 2023*). In particular, F-actin and Phospho-Ser PKC substrate acquired in magenta were changed to blue (*Figure 4—figure supplement 3*, *Figure 4—figure supplement 4*) and, besides that, YFP acquired in yellow was changed to green (i.e. *Figure 8* and *Figure 8—figure supplement 1*). Profile analyses of MFI corresponding to each separate channel and the colocalization pixels at the IS interface (ZX axes) were performed by using 'Intensity Profile' tool in NIS-AR and then Excel, as reported (*Ruiz-Navarro et al., 2023*). To measure the relative size of the F-actin low area at the cIS, which quantifies the decrease in F-actin density at the cIS, we used the last frame from the former IS interface videos (*Figure 7—video 1* and *Figure 7*), generated as described (*Ruiz-Navarro et al., 2023*). The areas of the F-actin-low region at cIS (Fact-low cIS area) (yellow line) and the synapse (IS area) (white line) (i.e. *Figure 7*) were defined in ImageJ using the ROI manager and applying an appropriate, manually defined threshold to the F-actin channel. The definition of the ROI to measure the Fact-low cIS area and the IS area was performed manually or by using automated algorithm 'auto-detect ROI/ segmentation' from NIS-AR (*Herranz et al., 2019*; *Bello-Gamboa et al., 2020*). As for the Fact-low cIS area (yellow line) and the IS area (white line) (i.e. *Figure 7*), they were defined in ImageJ using the ROI manager and applying a default threshold or, in cases where precise detection was not achieved, an appropriate, manually defined threshold to the F-actin channel. In both cases, we have found that both manual and automated values were quite similar. The areas included in these ROIs were measured and limited to the defined thresholds, and the relative area of the F-actin-low region at the cIS (Fact-low cIS area/IS area), which is independent of both cell and synapse size, was calculated and represented (*Figure 7*). Regarding the possibility that actin cytoskeleton of Raji cells can also contribute to the measurements of synaptic F-actin, it is important to remark that MHC-II-antigen triggering on the B cell side of the Th synapse does not induce noticeable F-actin changes along the synapse (i.e. F-actin clearing at the cIS), in contrast to TCR stimulation on T cell side (*Na et al., 2016*; *Yuseff et al., 2013*; *Calvo and Izquierdo, 2021*). In addition, we have observed that majority of F-actin at the IS belongs to the Jurkat cell (*Ruiz-Navarro et al., 2023*). Thus, the contribution to

the analyses of the residual, invariant F-actin from the B cell is negligible using our protocol (*Ruiz-Navarro et al., 2023*).

To compare MTOC and MVB polarization in synapses, MTOC and MVB PI were calculated, as previously described (*Obino et al., 2017*; *Herranz et al., 2019*; *Bello-Gamboa et al., 2020*; *Fernández-Hermira et al., 2023*; *Figure 4—figure supplement 1A*), using MIP, both for deconvoluted, widefield, and confocal-acquired images. In the MIP, the positions of the cell geometric center (Cell$^C$) and MTOC and MVB center of mass (MTOC$^C$ and MVB$^C$, respectively) were used to project MTOC$^C$ (or MVB$^C$) on the vector defined by the Cell$^C$-synapse axis. Then the MTOC (or MVB) PI was calculated by dividing the distance between the MTOC$^C$ (or MVB$^C$) projection and the Cell$^C$ ('B' or 'A' distance, respectively) by the distance between the Cell$^C$ and the synapse ('C' distance) (*Figure 4—figure supplement 1A*, *Figure 4*, and *Figure 5*). Cell$^C$ position was taken as the origin to measure distances, thus those 'A' and 'B' values in the opposite direction to the synapse were taken as negative. Thus, PI (PI = A/C or B/C) ranked from +1 (fully polarized) to –1 (fully anti-polarized). Therefore, PI values were normalized by cell size and shape (*Figure 4*; *Herranz et al., 2019*; *Fernández-Hermira et al., 2023*) and measured the relative ability of MTOC and MVB to polarize toward the IS. Remarkably, one important feature of the IS consists of both the onset of the initial cell-cell contacts and the establishment of a mature, fully productive IS are intrinsically stochastic, rapid, and asynchronous processes (*Yi et al., 2013*; *Friedl et al., 2005*; *Calvo and Izquierdo, 2018*). Thus, the score of the PI corresponding to the distance of MTOC/MVB with respect to the IS (*Kupfer and Singer, 1989*) may be contaminated by background MTOC/MVB polarization, in great part due to the stochastic nature of IS formation (*Yi et al., 2013*). In order to circumvent this caveat, a substantial number of IS of each cell group/condition were analyzed to obtain statistically significant results, as reported by numerous authors (*Duchez et al., 2011*; *Obino et al., 2017*; *Obino et al., 2016*; *Yi et al., 2013*; *Herranz et al., 2019*; *Bello-Gamboa et al., 2020*). Image analysis data correspond to at least three different experiments, analyzing a minimum of 30 synapses from 15 different, randomly selected, microscopy fields per experiment. ANOVA was performed for statistical significance of the results using Excel and IBM's SPSS Statistics software, followed by Tukey's post hoc analyses (provided as Source data).

## Acknowledgements

We are indebted and acknowledge Dr. DD Billadeau (Mayo Clinic, USA) for generous sharing of shFMNL1 and FMNL1 isoform rescue constructions. We acknowledge the excellent technical support from A Sánchez. We acknowledge Dr. MA Alonso (CBM, CSIC) for reagents and scientific advice. We acknowledge the generous contribution of pre-graduate students Alejandro Martín, Irene Gascuña, Gregorio Pantoja, María Ruiz, Sofía Blázquez, Carlos del Hoyo, Pablo del Barrio, and Elena Fernández to this work. Thanks to D Morales (SIDI-UAM), A Oña, and GD'Agostino (CNB, CSIC) and M Martín (IIBM,CSIC) for their superb expertise with confocal microscopy. Work in the F.R.G-G lab was funded by grant PID2019-104941RB-I00 from the Spanish Ministry of Science and Innovation (PID2019-104941RB-I00/AEI/10.13039/501100011033). This work was supported by a grant from the Programa Estatal de Investigación, Desarrollo e Innovación, Modalidad Retos Investigación (Grant PID2020-114148RB-I00) funded by Spanish Ministry of Science and Innovation (PID2020-114148RB-I00/AEI/10.13039/501100011033), and grant P2022/BMD-7225, funded by Consortia in Biomedicine of Comunidad de Madrid to MI.

## Additional information

### Funding

| Funder | Grant reference number | Author |
|---|---|---|
| Spanish Ministry of Science and Innovation | PID2020-114148RB-I00/AEI/10.13039/501100011033 | Manuel Izquierdo Pastor |
| Spanish Ministry of Science and Innovation | PID2019-104941RB-I00/AEI/10.13039/501100011033 | Francesc R Garcia-Gonzalo |
| Comunidad de Madrid | P2022/BMD-7225 | Manuel Izquierdo Pastor |

| Funder | Grant reference number | Author |
|---|---|---|

The funders had no role in study design, data collection and interpretation, or the decision to submit the work for publication.

## Author contributions

Javier Ruiz-Navarro, Data curation, Software, Formal analysis, Validation, Investigation, Visualization, Methodology, Writing – original draft, Writing – review and editing; Sara Fernández-Hermira, Formal analysis, Investigation, Methodology; Irene Sanz-Fernández, Pablo Barbeito, Alfonso Navarro-Zapata, Investigation, Methodology; Antonio Pérez-Martínez, Resources, Investigation, Methodology; Francesc R Garcia-Gonzalo, Resources, Investigation, Methodology, Writing – original draft; Víctor Calvo, Conceptualization, Data curation, Formal analysis, Supervision, Validation, Investigation, Visualization, Methodology, Writing – original draft, Writing – review and editing; Manuel Izquierdo Pastor, Conceptualization, Data curation, Formal analysis, Supervision, Funding acquisition, Validation, Investigation, Visualization, Methodology, Writing – original draft, Project administration, Writing – review and editing

## Author ORCIDs

Javier Ruiz-Navarro http://orcid.org/0009-0005-8393-0450
Pablo Barbeito https://orcid.org/0000-0003-0758-0012
Francesc R Garcia-Gonzalo https://orcid.org/0000-0002-9152-2191
Manuel Izquierdo Pastor https://orcid.org/0000-0002-7701-1002

Joint Public Review: https://doi.org/10.7554/eLife.96942.4.sa1
Author response https://doi.org/10.7554/eLife.96942.4.sa2

# Additional files

## Supplementary files

• MDAR checklist

• Source data 1. Tukey's post hoc analyses corresponding to all dot plot figures.

## Data availability

We declare we have deposited our Source data (Excel dataset) corresponding to all the dot plot data presented in this contribution with Dryad, a generic public repositorie, and thus the materials described in the manuscript, including all relevant raw data, will be freely available to any researcher wishing to use them for non-commercial purposes, without breaching participant confidentiality. Supporting files: source data files have been provided for Figures 2, 3, 6 and 9, containing images of WB blots that are provided as original Source data of uncropped, unedited WB, associated to each corresponding figure. Both PDF of WB and original TIF files are provided. In addition, *Source data 1* contains p-values after applying Tukey's method (POST HOC) to the one-way ANOVA of dot plots from Figures 4, 5 and 7 are provided.

The following dataset was generated:

| Author(s) | Year | Dataset title | Dataset URL | Database and Identifier |
|---|---|---|---|---|
| Pastor et al. | 2024 | Formin-like 1 β phosphorylation at S1086 is necessary for secretory polarized traffic of exosomes at the immune synapse in Jurkat T lymphocytes | https://doi.org/10.5061/dryad.j9kd51cnf. | Dryad Digital Repository, 10.5061/dryad.j9kd51cnf |

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
