## [Editor Report · eLife Assessment]

This **important** study uses the Jurkat T cell model to study the role of Formin-like 1 β phosphorylation at S1086 on actin dynamics and exosome release at the immunological synapse. The evidence supporting these findings is **compelling** within the framework of the Jurkat model. As the Jurkat model is known to have a bias toward formin-mediated actin filament formation at the expense of Arp2/3-mediated branched F-actin foci observed in primary T cells, it will be beneficial in the future to confirm major findings in primary T cells.

---

## [Referee Report · Joint Public Review]

Summary

Based on (i) the documented role of FMNL1 proteins in IS formation; (ii) their ability to regulate F-actin dynamics; (iii) the implication of PKCdelta in MVB polarization to the IS and FMNL1beta phosphorylation; and (iv) the homology of the C-terminal DAD domain of FMNL1beta with FMNL2, where a phosphorylatable serine residue regulating its auto-inhibitory function had been previously identified, the authors have addressed the role of S1086 in the FMNL1beta DAD domain in F-actin dynamics, MVB polarization and exosome secretion, and investigated the potential implication of PKCdelta, which they had previously shown to regulate these processes, in FMNL1beta S1086 phosphorylation. They demonstrate that FMNL1beta is indeed phosphorylated on S1086 in a PKCdelta-dependent manner and that S1086-phosphorylated FMNL1beta acts downstream of PKCdelta to regulate centrosome and MVB polarization to the IS and exosome release. They provide evidence that FMNL1beta accumulates at the IS where it promotes F-actin clearance from the IS center, thus allowing for MVB secretion.

Strengths

The work is based on a solid rationale, which includes previous findings by the authors establishing a link between PKCdelta, FMNL1beta phosphorylation, synaptic F-actin clearance and MVB polarization to the IS. The authors have thoroughly addressed the working hypotheses using robust tools. Among these, of particular value is an expression vector that allows for simultaneous RNAi-based knockdown of the endogenous protein of interest (here all FMNL1 isoforms) and expression of wild-type or mutated versions of the protein as YFP-tagged proteins to facilitate imaging studies. The imaging analyses, which are the core of the manuscript, have been complemented by immunoblot and immunoprecipitation studies, as well as by the measurement of exosome release (using a transfected MVB/exosome reporter to discriminate exosomes secreted by T cells).

Weaknesses

As stated in the title of the article, the main findings have been obtained in clones of Jurkat cells and have not been confirmed in primary T cells.

---

## [Author Response]

The following is the authors’ response to the previous reviews.

**eLife Assessment**
This is a valuable study in the Jurkat T cell line that calls attention to phosphorylation of formin-like 1 β role and its role in polarization of CD63 positive extracellular vesicles (referred to as exosomes). The evidence presented in the Jurkat model is solid, but concerns have been raised about the statistical analysis and more details would be required to fully assess the significance of the results. For example, ANOVA is the method described, but it requires large amounts of normally distributed data in multiple groups and cannot be used to make pairwise comparisons within groups, which would require a post-hoc method (which is not discussed). In addition, the data showing forming-like 1 β in primary human T cells without and with a CAR are provided without quantification and don't investigate any of the novel claims, so doesn't address the relevance of Formin-like 1 β beyond the Jurkat model. Nonetheless, the consistent trends in the body of the study provide solid support for the claims.

We acknowledge this general statement on statistics. Thus, we have now discussed and provided more details on the post-hoc method (Tukey), as a new Supplementary data S13 (p-values after applying tukey's method -post hoc- to the one-way anova for all the pairwise comparisons). Additionally, we have now provided quantitative data on the percentage of primary cells with and without CAR that show FMNL1 accumulations at the immune synapse (Suppl. Fig. S7). Regarding the data in primary human T cells, we have already changed the title of the manuscript to strictly adjust it to the main body of the data and our conclusions in the well-established Jurkat synapse model. We also want to emphasize that we have not pretended to extrapolate the relevance of our data regarding FMNL1 and exosomes beyond the Jurkat model. Thus, we have included some additional sentences and/or nuances in the Discussion to somewhat soften our statements in this regard (i.e. “…..provided that the FMNL1 effect on exosome secretion in Jurkat cells can be extended to primary T lymphocytes”) and to clarify this important point.

**Reviewer 1:**
(1) The main findings have been obtained in clones of Jurkat cells. They have not been confirmed in primary T cells. The only experiment performed in primary cells is shown in Figure S7 (primary human T lymphoblasts) for which only the distribution of FMNL1 is shown without quantification. No results presenting the effect of FMNL1 KO and expression of mutants in primary T cells are shown.

Referee is right regarding the extension of exosome secretion studies to primary human T lymphocytes. Unfortunately, it is well known that primary T lymphocytes are extremely difficult to transfect. Moreover, the expression of our large bi-cistronic large plasmids (>15 Kb) is very inefficient, coupled with the challenge of expressing large proteins, such as the 180 kDa YFP-FMNL1 chimeric variants. The convergence of all these undesirable factors synergistically hampers these studies and we have been unable to consistently achieve enough transfection efficiency to perform these experiments. However, the role of FMNL1 on MTOC/MVB polarization in Jurkat cells, confirmed in this manuscript, has been already extended to primary CD8+ T cell clones (DOI10.1016/j.immuni.2007.01.008). Given that exosome secretion requires

MTOC/MVB polarization both in Jurkat and primary T lymphoblasts (10.1038/cdd.2010.184, 10.3389/fimmu.2019.00851), this suggests FMNL1 may also control exosome secretion in primary T cells, although the formal demonstration will require further research.

A new sentence has been included in the Discussion to address this important point. Regarding the second request, we have quantified the images mentioned in Suppl. Fig. S7, and the percentages of fixed T cells showing FMNL1 accumulations at the immune synapse are included in the figure legend.

(2) Analysis in- depth of the defect in actin remodeling (quantification of the images, analysis of some key actors of actin remodeling) is still lacking. Only Factin is shown, no attempt to look more precisely at actors of actin remodeling has been done.

The referee is right. Since we have obtained new results on the role of FMNL1 on actin remodeling, we have focused on this formin, which is already a key actor in this process. In this context, we have previously shown that the formin Dia1, another major actor of actin remodeling in T lymphocytes along with FMNL1 (DOI10.1016/j.immuni.2007.01.008), does not undergo phosphorylation upon PKC activation (Suppl. Fig. 5 in https://doi.org/10.1080/20013078.2020.1759926). Since our aim was to unravel the PKC-mediated pathway controlling actin remodeling, we have ruled out more studies on Dia1. Therefore, we have included a new sentence to emphasize the specific role of FMNL1 phosphorylation, but not Dia1, in this regard. Nonetheless, future studies aimed to identifying new important players in this or related pathways could offer significant insights.

(3) The defect in the secretion of extracellular vesicles is still very preliminary. Examples of STED images given by the authors are nice, yet no quantification is performed.

The referee is right regarding this point and we acknowledge this comment. Accordingly, we have now quantified the STED images and provided numerical data on the percentages of cells exhibiting the observed phenotypes (see the figure legend for Fig. 10).

(4) Results shown in Figure S12 on the colocalization of proteins phosphorylated on Ser/Thr are still not convincing. It seems indeed that "phospho-PKC" is labeling more preferentially the CMAC positive cells (Raji) than the Jurkat T cells. It is thus particularly difficult to conclude on the colocalization and even more on the recruitment of phosphorylated-FMNL1 at the IS. Thus, these experiments are not conclusive and cannot be the basis even for their cautious conclusion: "Although all these data did not allow us to infer that FMNL1b is phosphorylated at the IS due to the resolution limit of confocal and STED microscopes, the results are compatible with the idea that both endogenous FMNL1 and YFP-FMNL1bWT are specifically phosphorylated at the cIS".

The referee may be correct regarding the detail of the "phospho-PKC" labeling. However, it cannot be overlooked that Raji cells also contain proteins that are or may be potential PKC substrates. As a matter of fact, Raji cells also express FMNL1. In addition, MHCII triggering in B cells induces PKC activation (https://doi.org/10.1002/eji.200323351). Regarding which cell type is preferentially labeled, this is a variable topic depending on the analyzed synapse.

It is true that there are likely several PKC substrates, both in Jurkat in Raji cells, but our point is that one of these substrates either colocalizes with FMNL1 or is FMNL1 itself. We do not claim at any point that FMNL1 is the only PKC substrate, neither in Jurkat or in Raji cells.

Apparently, the referee has either overlooked our results or we did not emphasize them sufficiently. Our results effectively validated the PKC substrate antibody, both on endogenous phospho-FMNL1 and phospho-YFPFMNL1β by WB (Fig. 3). Moreover, the phospho-PKC does not recognize

YFP-FMNL1β S1086A or S1086D variants (Fig. 3). Last, but not least, when FMNL1 is interfered in the Jurkat cell, the phospho-PKC does not colocalize with FMNL1, but it strongly colocalizes at the synapse with expressed YFPFMNL1βWT in the Jurkat cell (Fig. S11). Indeed YFP-FMNL1β belonged to the Jurkat cell. Taken together these results demonstrate: 1. the specificity of phospho-PKC antibody, 2. the phospho-PKC antibody certainly recognizes phosphorylated YFP-FMNL1β but not its non-phosphorylatable mutant variants, 3. the colocalization of phospho-PKC with anti-FMNL1 is specific. We have included some sentences to clarify these points and to avoid possible misunderstandings by potential readers. We acknowledge the referee for his/her clarifying point, and we firmly believe our mentioned cautious conclusion is strictly correct, although we have tuned it to consider the possibility that a different PKC substrate could be closely associated to FMNL1, producing the observed colocalization: “Although all these data do not yet allow us to infer that FMNL1b is phosphorylated at the IS due to the resolution limits of super resolution microscopy and the possibility that another PKC substrate may be associated to FMNL1 or very close to FMNL1, in a strictly S1086-dependent manner”.

To clear any doubt regarding which cell is labelled with phospho-PKC, we have changed the lower panels in Suppl. Fig. S12, and now is more evident that FMNL1 and phospho-PKC belong to the Jurkat cell.

The study would benefit from a more careful statistical analysis. The dot plots showing polarity are presented for one experiment. Yet, the distribution of the polarity is broad. Results of the 3 independent experiments should be shown and a statistical analysis performed on the independent experiments.

The referee is right and we have now included further post-hoc analyses data (Tukey) at Suppl. Fig S13. Tukey’s test values were included for all the dot plot figures. We have not included all the plots from 3 different experiments since the manuscript already contains 10+12 multi panel figures and is too large. However, we have stated in the figure legend that these independent experiments are representative of the data obtained from 3 independent experiments. Referee’s consideration regarding the broad distribution of polarity data is correct. We included in the first version of the manuscript a sentence in this regard, that it may have been overlooked: “Remarkably, one important feature of the IS consists of both the onset of the initial cell-cell contacts and the establishment of a mature, fully productive IS, are intrinsically stochastic, rapid and asynchronous processes (87, 88) (43). Thus, the score of the PI corresponding to the distance of MTOC/MVB with respect the IS (42) may be contaminated by background MTOC/MVB polarization, in great part due to the stochastic nature of IS formation (87)”.